# LANGUAGE MODEL EMPOWERED SPATIO-TEMPORAL FORECASTING VIA PHYSICS-AWARE REPROGRAMMING

## ABSTRACT

Spatio-temporal forecasting is pivotal in numerous real-world applications, including transportation planning, energy management, and climate monitoring. In this work, we aim to harness the reasoning and generalization abilities of Pre-trained Language Models (PLMs) for more effective spatio-temporal forecasting, particularly in data-scarce scenarios. However, recent studies uncover that PLMs, which are primarily trained on textual data, often falter when tasked with modeling the intricate correlations in numerical time series, thereby limiting their effectiveness in comprehending spatio-temporal data. To bridge the gap, we propose REPST, a physics-aware PLM reprogramming framework tailored for spatio-temporal forecasting. Specifically, we first propose a physics-aware decomposer that adaptively disentangles spatially correlated time series into interpretable sub-components, which facilitates PLM to understand sophisticated spatio-temporal dynamics via a divide-and-conquer strategy. Moreover, we propose a selective discrete reprogramming scheme, which introduces an expanded spatio-temporal vocabulary space to project spatio-temporal series into discrete representations. This scheme minimizes the information loss during reprogramming and enriches the representations derived by PLMs. Extensive experiments on real-world datasets show that the proposed REPST outperforms twelve state-of-the-art baseline methods, particularly in data-scarce scenarios, highlighting the effectiveness and superior generalization capabilities of PLMs for spatio-temporal forecasting.

## 1 INTRODUCTION

Spatio-temporal forecasting aims to predict future states of real-world complex systems by simultaneously learning spatial and temporal dependencies of historical observations, which plays a pivotal role in diverse real-world applications, such as traffic management (Li et al., 2018; Wu et al., 2019), environmental monitoring (Han et al., 2023), and resource optimization (Geng et al., 2019). In the past decade, deep learning has demonstrated great predictive power and led to a surge in deep spatio-temporal forecasting models (Xie et al., 2020; Jin et al., 2023a). For example, Recurrent Neural Networks (RNNs) and Graph Neural Networks (GNNs) are frequently combined to capture complex patterns for spatio-temporal forecasting (Jin et al., 2023a; Li et al., 2018; Han et al., 2020). Despite fruitful progress made so far, such approaches are typically confined to the one-task-one-model setting, which lacks general-purpose utility and inevitably falls short in handling widespread data-scarcity issue in real-world scenarios, *e.g.*, newly deployed monitoring services.

In recent years, Pre-trained Language Models (PLMs) like GPT-3 (Brown, 2020) and the LLaMA family (Touvron et al., 2023) have achieved groundbreaking success in the Natural Language Processing (NLP) domain. PLMs exhibit exceptional contextual understanding, reasoning, and few-shot generalization capabilities across a wide range of tasks due to their pre-training on extensive text corpora. Although originally designed for textual data, the versatility and power of PLMs have inspired their application to numerically correlated data (Zhou et al., 2024; Jin et al., 2023b; 2024). For example, Frozen Pretrained Transformer (FPT) (Zhou et al., 2024) pioneers research in this direction and showcases the promise of fine-tuning PLMs as generic time series feature extractors. Besides, model reprogramming (Jin et al., 2023b; 2024) have considered the modality differences between time series and natural language, solving time series forecasting tasks by learning an input transformation function that maps time series patches (Nie et al., 2022) to a compressed vocabulary.

In this work, we aim to harness the reasoning and generalization abilities of PLMs for more effective spatio-temporal forecasting, particularly in data-scarce environments.

However, significant challenges remain in directly applying aforementioned reprogramming techniques to spatio-temporal forecasting. The foremost issue lies in the underutilization of PLMs' full potential. Recent work (Tan et al., 2024) suggests that existing PLM-based approaches for time series forecasting (Zhou et al., 2024; Jin et al., 2023b; Wang et al., 2024) fail to leverage the generative and reasoning abilities of PLMs. This limitation becomes even more apparent when handling more complex spatially correlated time series data. A crucial question that arises is: *how can we better explain this shortcoming and unlock the potential of PLMs for spatio-temporal forecasting?* Another challenge is PLMs' limited capacity to model the intricate correlations present in spatio-temporal data. While PLMs excel at capturing dependencies within one-dimensional sequential data, they fall short in comprehending spatio-temporal data, which often has more complex structures like grids or graphs (Li et al., 2024c). This gap in PLMs' understanding of spatio-temporal correlations poses a significant obstacle to their effective use in this domain.

To address these challenges, we argue that the primary limitation of existing approaches lies in their oversimplified treatment of spatio-temporal data, which prevents PLMs from fully understanding the underlying semantics. Instead of merely serving as a one-dimensional encoder, PLMs need a more sophisticated understanding to handle spatio-temporal data effectively. Through explorative experimental study, we observe even apply simple decomposition techniques can significantly facilitate PLMs to better understand spatio-temporal data and leading to improved performance, as shown in Figure 1.

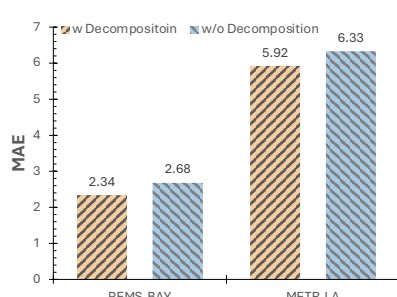

Figure 1: Simple Fourier-based decomposition (Liu et al., 2024c) can improve the PLM's understanding of spatio-temporal data. We report the forecasting results by applying reprogrammed GPT-2 (Jin et al., 2023b) on widely used PEMS-BAY and METR-LA (Li et al., 2018) datasets.

Building on this insight, we propose **REPST**, a reprogramming framework specifically designed for spatio-temporal forecasting using PLMs. Specifically, we first propose a **physics-aware spatio-temporal decomposer**, which adaptively disentangles spatio-temporal dynamics into components that represent physical processes within the system. This is achieved through a Koopman theory-based evolutionary matrix, which results in decomposed components rich in spatio-temporal semantics that PLMs can more easily comprehend. This decomposition-based approach enables PLMs to capture both spatial and temporal dynamics more effectively. Moreover, we introduce a **selective reprogramming strategy** to tackle the complexity of spatio-temporal structures, which differ fundamentally from the one-dimensional sequence-like structure of textual data. Our strategy constructs an expanded spatio-temporal vocabulary by selecting the most relevant spatio-temporal word tokens from the PLM's vocabulary through a differentiable reparameterization process. Unlike previous works that use compressed vocabularies, which can lead to ambiguous semantics, our approach reconstructs the reprogramming space with a rich, semantically distinct spatio-temporal vocabulary. By leveraging pretrained spatio-temporal correlations, this strategy enables PLMs to focus on relationships among tokens in a 3D geometric space, significantly enhancing their ability to model complex spatio-temporal dynamics. We evaluate REPST on a variety of spatio-temporal forecasting tasks, including energy management, air quality prediction, and traffic forecasting. Extensive experimental results highlight the framework's superior performance compared to state-of-the-art models, particularly in few-shot and zero-shot learning contexts. Our main contributions are summarized as:

- We identify the underlying reason for the underperformance of existing PLM-based approaches for spatio-temporal forecasting, highlighting the need to decompose spatio-temporal dynamics into interpretable components to fully leverage PLMs' potential.

- We propose REPST, a spatio-temporal forecasting framework that enables PLMs to grasp complex spatio-temporal patterns via physics-aware decomposition-based reprogramming. The reprogramming module reconstructs an expanded spatio-temporal vocabulary using a selective strategy, allowing PLMs to model spatio-temporal dynamics without altering their pre-trained parameters.

- We show that REPST consistently achieves superior performance across real-world datasets, particularly in data-scarce settings, demonstrating strong generalization capabilities in few-shot and zero-shot learning scenarios.

## 2 PRELIMINARIES

Spatio-temporal data can be considered as observations of the state of a dynamical system. It is typically represented as a two-dimensional matrix $\mathbf{X} \in \mathbb{R}^{N \times T}$, which captures the states of a set of N nodes $\mathcal{V}$, where each node in $\mathcal{V}$ corresponds to an entity (*e.g.*, grids, regions, and sensors) in space. Specifically, we denote $\mathbf{x}_{t-T+1:t}^i = [\mathbf{x}_{t-T+1}^i, \mathbf{x}_{t-T}^i, ..., \mathbf{x}_t^i]^\top \in \mathbb{R}^{T \times 1}$ as the observations of node $i$ from time step $t - T + 1$ to $t$, where $T$ represents the look-back window length. The goal of spatio-temporal forecasting problem is to predict future states for all nodes $i \in \mathcal{V}$ over the next $\tau$ time steps based on a sequence of historical observations. This involves uncovering the complex spatial and temporal patterns inherent in spatio-temporal data to reveal the hidden principles governing the system's dynamics:

$$\hat{\mathbf{Y}}_{t+1:t+\tau} = f_\theta(\mathbf{X}_{t-T+1:t}), \tag{1}$$

where $\mathbf{X}_{t-T+1:t} = [\mathbf{x}_{t-T+1:t}^0, \mathbf{x}_{t-T+1:t}^1, ..., \mathbf{x}_{t-T+1:t}^{N-1}]^\top \in \mathbb{R}^{N \times T}$ denotes the historical observations in previous $T$ time steps, and $f_\theta(\cdot)$ is the spatio-temporal forecasting model parameterized by $\theta$. $\hat{\mathbf{Y}}_{t+1:t+\tau} = \{\hat{\mathbf{y}}_{t+1:t+\tau}^i\}_{i=0}^N$ and $\mathbf{Y}_{t+1:t+\tau} = \{\mathbf{y}_{t+1:t+\tau}^i\}_{i=0}^N$ denote the estimated future states and the ground truth in the next $\tau$ time steps, where $\hat{\mathbf{Y}}_{t+1:t+\tau}, \mathbf{Y}_{t+1:t+\tau} \in \mathbb{R}^{N \times \tau}$. For convenience, we omit the lower corner mark and represent $\mathbf{X}_{t-T+1:t}, \mathbf{Y}_{t+1:t+\tau}$ as $\mathbf{X}, \mathbf{Y}$ and $\mathbf{x}_{t-T+1:t}^i, \mathbf{y}_{t+1:t+\tau}^i$ as $\mathbf{x}^i, \mathbf{y}^i$.

## 3 METHODOLOGY

As illustrated in Figure 2, REPST consists of three components: a physics-aware spatio-temporal decomposer, a selective reprogrammed language model, and a learnable mapping function. To be specific, in physics-aware spatio-temporal decomposer, we first decompose the input signals into a series of distinct physics-aware components. Then, we utilize an adaptive reprogramming strategy to reprogram these components into textual embeddings via an expanded spatio-temporal vocabulary constructing through a selective manner. After that, we employ a frozen PLM to construct spatio-temporal correlations based on these textual embeddings. Finally, a learnable mapping function generates future predictions based on the output of PLM.

### 3.1 PHYSICS-AWARE EVOLUTIONAL SPATIO-TEMPORAL DECOMPOSITION

Recent studies have revealed that PLMs possess rich spatio-temporal knowledge and reasoning capabilities (Gurnee & Tegmark, 2023; Mai et al., 2023; Jin et al., 2024). However, existing methods failed to fully leverage the capabilities of PLMs, which raises challenges for spatio-temporal data forecasting as well. As aforementioned, reasons for this shortcoming lies in their over simplistic encoding to time series. PLMs requires further process of spatio-temporal data to enhance their comprehensibility to such complex structure. In this section, we address this shortcoming through a carefully designed physics-aware spatio-temporal decomposer. Previous works (Liu et al., 2024c; Yi et al., 2024; Shao et al., 2022b) decouple the time series in Fourier space and handle the decoupled signals separately for better use of the hidden information of time series. Simply decomposing time series solely based on frequency intensity is not interpretable and cannot effectively capture the highly coupled spatio-temporal dynamics. Furthermore, this cannot be easily realized by language models as well due to their limited comprehension of physical semantics.

To fully unlock the spatio-temporal knowledge, inspired by dynamic mode decomposition (Schmid, 2010; Kutz et al., 2016), we propose to capture the underlying dynamic signals in an interpretable manner by leveraging the dynamic system's evolution matrix $\mathcal{A}$. To be specific, considering two state observations $\mathbf{X}_{1:t-1}$ and $\mathbf{X}_{2:t}$, it satisfies $\mathbf{X}_{2:t} = \mathcal{A}\mathbf{X}_{1:t-1}$. This evolution matrix $\mathcal{A}$ is sought in a low-rank setting to capture the modes governing the system's dynamics. By applying a series of mathematical process such as singular value decomposition (SVD) to $\mathbf{X}_{1:t-1}$ and $\mathbf{X}_{2:t}$, we obtain the eigenvectors $\Omega = [\omega_1, \omega_2, ..., \omega_C]$ and corresponding eigenvalues $V = [v_1, v_2, \dots, v_C]$, which can be leveraged to decompose spatio-temporal dynamic systems into different components. Each $\omega_i$, referred to as a mode of the dynamical system, reflects certain physical behaviors of the system. We provide a detailed calculation process in Appendix A.5.

Specifically, we first obtain $\mathbf{X}_{norm}$ by normalizing the input $\mathbf{X}$ for each node to have zero mean and unit standard deviation using reversible instance normalization (RevIN) (Kim et al., 2021).

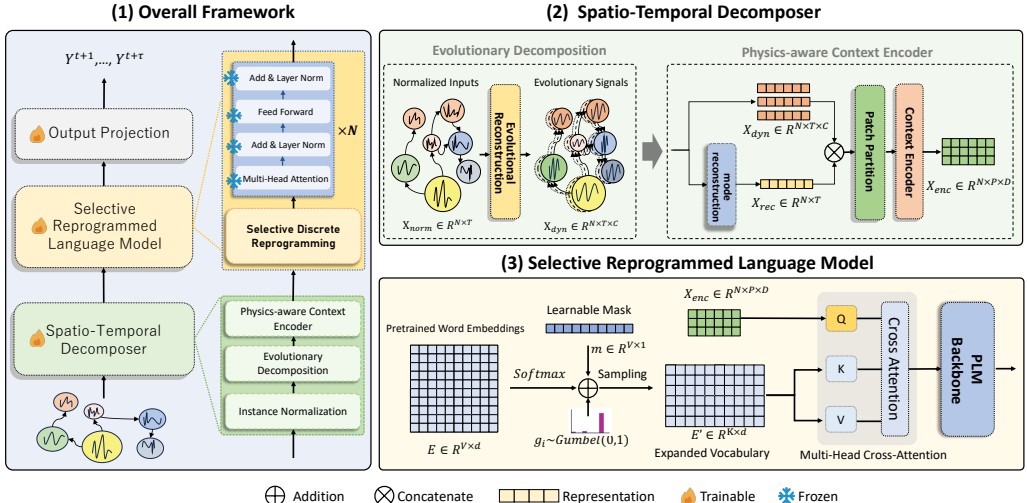

Figure 2: The model framework of REPST. (1) Given a raw input spatio-temporal data, we first perform normalization and then decouple the spatio-temporal data into a set of evolutionary signals. (2) After that, the signals are concatenated and divided into patches and further transformed into embeddings by a physics-aware context encoder. (3) Then, the patch embeddings are aligned with natural language by reprogramming with expanded spatio-temporal vocabulary and further processed by the frozen GPT-2 backbone. The output patches of the pre-trained language model are reprocessed by a learnable mapping function to generate the forecasts.

Then, we disentangle a set of physics-aware dynamic components $\mathbf{X}_{dyn} \in \mathbb{R}^{N \times T \times C}$ from intricate spatio-temporal data through reconstructing the system dynamics via modes $\omega_i$ from the system's evolution matrix's eigenvectors $\Omega$ and the corresponding eigenvalues $v_i$. By explicitly decoupling the physics-aware nature of the spatio-temporal system, our approach is well-suited to capture the various physical behaviors of the system, providing PLMs with a series of components enriched with spatio-temporal semantic information that is significantly easier to comprehend compared to the originally densely coupled dynamic signals (Rowley et al., 2009; Li et al., 2024a).

$$\mathbf{X}_{dyn} = \varepsilon_0 e^{\omega_0 t} v_0 \parallel \varepsilon_1 e^{\omega_1 t} v_1 \parallel \cdots \parallel \varepsilon_C e^{\omega_C t} v_C, \tag{2}$$

where $\mathbf{X}_{dyn}$ is a set of spatio-temporal dynamics calculating based on the modes $\omega_i$ and eigenvalues $v_i$. $\varepsilon_i$ is based on the input observation (see section A.5). Since the dynamics of the system is disentangled, we can distinguish the noise from the dominant dynamic signals in the original data. If only the most significant information is retained during reconstruction, the reconstruction results can remove noise, thus obtaining a smoother state evolutionary information. Therefore, we further reconstruct the whole system $\mathbf{X}_{rec} \in \mathbb{R}^{N \times T}$ with most dominant modes to enhance prediction:

$$\mathbf{X}_{rec} = \sum_i \varepsilon_i e^{\omega_i t} v_i, i \in \alpha, \tag{3}$$

where $\alpha$ represents a set of indices stands for top-k most dominant modes, constructed based on each mode's contribution to the overall system (Schmid, 2010), which is calculated through the analysis of $\omega_i$ and $v_i$ (see section A.5). Compared to existing Fourier-based methods, the system's evolution matrix $\Omega$ is derived from data representing the true dynamics of the system. It can separate modes corresponding to specific physical processes, enabling us to capture various aspects of the system's evolution, such as periodic oscillations in traffic flows caused by traffic signals or slow changes in air pollution driven by wind direction (Proctor et al., 2016; Brunton et al., 2016; Chen et al., 2012).

Additionally, to enhance the information density of decoupled signals, we employ patching strategy (Nie et al., 2022) to construct patches as the input tokens for PLMs. Given the decoupled signals $\mathbf{X}_{dec} = \mathbf{X}_{rec} \parallel \mathbf{X}_{dyn} \in \mathbb{R}^{N \times T \times (C+1)}$, we divide the observations of each node as a series of non-overlapped patches $\mathbf{X}_{dec}^P \in \mathbb{R}^{N \times P \times T_P \times (C+1)}$, where $P = [T/T_P] + 1$ represents the number of the resulting patches, and $T_P$ denotes the patch length. Next, we encode the patched signals as patched embeddings : $\mathbf{X}_{enc} = \text{Conv}(\mathbf{X}_{dec}^P, \theta_p) \in \mathbb{R}^{N \times P \times D}$, where $N$ stands for the number of nodes, and $D$ is the embedding dimension. $\text{Conv}(\cdot)$ denotes the patch-wise convolution operator

and $\theta_p$ represents the learnable parameters of the patch-wise convolution. Unlike previous works (Liu et al., 2024a;b) that simply regard each node as a token, our model treats each patch as one token, allowing to construct fine-grained relationships among both spatial and temporal patterns. By doing so, our model can preserve representations rich in semantic information, allowing PLM's comprehension in both spatial and temporal dynamics more effectively.

## 3.2 SELECTIVE REPROGRAMMED LANGUAGE MODELS

Based on the decoupled signal patches $\mathbf{X}_{enc}$, how to tackle the complexity of spatio-temporal structures raises another question. Compared to directly handling the spatio-temporal embeddings, representations in natural language space are inherently suitable for PLMs. To enrich spatio-temporal semantics and enable more comprehensive modeling of hidden spatio-temporal physical relationships, as well as unlocking the reasoning capabilities of PLMs, we further reprogram the components into the textual embedding place via an expanded spatio-temporal vocabulary. When handling textual-based components, the rich physical semantic information can boost the pretrained physical knowledge of PLMs, resulting in an adequate modeling of the hidden physical interactions between disentangled components.

Specifically, we introduce our selective reprogramming strategy, which further constructs an expanded spatio-temporal vocabulary in a differentiable reparameterization process. We begin with $\mathbf{E} \in \mathbb{R}^{V \times d}$, the pretrained vocabulary of the PLMs, where $V$ is the vocabulary size and $d$ is the dimension of the embedding. We introduce a learnable word mask vector $\mathbf{m} \in \mathbb{R}^{V \times 1}$ to adaptively select the most relevant words, where $\mathbf{m}[i] \in \{0, 1\}$. In specific, we first obtain $\mathbf{m}$ through a linear layer followed by a Softmax activation, denoted as $\mathbf{m} = \text{Softmax}(\mathbf{EW})$, where $\mathbf{W}$ is a learnable matrix. Afterward, we sample Top-K word embeddings from $\mathbf{E}$ based on probability $\mathbf{m}[i]$ associated with word $i$ for reprogramming. Since the sampling process is non-differentiable, we employ Gumbel-Softmax trick(Jang et al., 2016) to enable gradient calculation with back-propagation, defined as

$$\mathbf{m}'[i] = \frac{\exp((\log \mathbf{m}[i] + g_i)/\tau)}{\sum_{j=1}^{V} \exp((\log \mathbf{m}[i] + g_j)/\tau)}, \tag{4}$$

where $\mathbf{m}'$ is a continuous relaxation of binary mask vector $\mathbf{m}$ for word selection, $\tau$ is temperature coefficient, $g_i$ and $g_j$ are i.i.d random variables sampled from distribution $\text{Gumbel}(0, 1)$. Concretely, the Gumbel distribution can be derived by first sampling $u \sim \text{Uniform}(0, 1)$ and then computing $g_i = -\log(-\log(u))$. By doing so, we can expand vocabulary space while preserving the semantic meaning of each word.

After obtaining the sampled word embeddings $\mathbf{E}' \in \mathbb{R}^{K \times d}$, we perform modality alignment by using cross-attention. In particular, we define the query matrix $\mathbf{X}_q = \mathbf{X}_{enc}\mathbf{W}_q$, key matrix $\mathbf{X}_k = \mathbf{E}'\mathbf{W}_k$ and value matrix $\mathbf{X}_v = \mathbf{E}'\mathbf{W}_v$, where $\mathbf{W}_q$, $\mathbf{W}_k$, and $\mathbf{W}_v$. After that, we calculate the reprogrammed patch embedding as follows:

$$\mathbf{Z} = \text{Attn}(\mathbf{X}_q, \mathbf{X}_k, \mathbf{X}_v) = \text{Softmax}(\frac{\mathbf{X}_q\mathbf{X}_k^\top}{\sqrt{d}})\mathbf{X}_v, \tag{5}$$

where $\mathbf{Z} \in \mathbb{R}^{N \times P \times d}$ denotes the aligned textual representations for the input spatio-temporal data. Based on the aligned representation, we utilize the frozen PLMs as the backbone for further processing. Roughly, PLMs consist of three components: self-attention, Feedforward Neural Networks, and layer normalization layer, which contain most of the learned semantic knowledge from pre-training. The reprogrammed patch embedding $\mathbf{Z}$ is encoded by this frozen language model to further process the semantic information and generates hidden textual representations $\mathbf{Z}_{text}$. A learnable mapping function is then used to generate the desired target outputs, which map the textual representations into feature prediction.

Overall, in our REPST, the process of predicting the future states $\hat{\mathbf{Y}}$ based on the history observation $\mathbf{X}$ can be simply formulated as:

$$\mathbf{Z} = \text{Evolutionary-Decomposition}(\mathbf{X}), \tag{6}$$

$$\mathbf{Z}_{text} = \text{Reprogrammed-LM}(\mathbf{Z}, \mathbf{E}), \tag{7}$$

$$\hat{\mathbf{Y}} = \text{Projection}(\mathbf{Z}_{text}), \tag{8}$$

where Evolutionary-Decomposition($\cdot$) represents the physics-aware spatio-temporal decomposer and Reprogrammed-LM($\cdot, \cdot$), Projection($\cdot$) is the selective reprogrammed Language Model and learnable mapping function.

**Model optimization.** Following the previous GNN-based works (Wu et al., 2019; Shao et al., 2022a), our REPST aims to minimize the mean absolute error (MAE) between the predicted future states $\hat{\mathbf{Y}}$ and ground truth $\mathbf{Y}$. This provides us effective capability to generate predictions among various spatio-temporal scenarios, formulated as: $\mathcal{L} = \frac{1}{N} \sum_{i=1}^{N} \left| \hat{\mathbf{y}}^i - \mathbf{y}^i \right|$ . Here, $\hat{\mathbf{y}}^i, \mathbf{y}^i$ represents a sample from $\hat{\mathbf{Y}}$ and $\mathbf{Y}$, and N represent the total number of samples.

**Scalability.** Technically, the proposed REPST can be viewed as utilizing GPT-2 after performing cross-attention over $N \times P$ patches and $V'$ word embeddings, which has the time and memory complexities that scale with $\mathcal{O}(N \cdot P \cdot V' + (N \cdot P)^2)$. Notably, the frozen GPT-2 blocks account for $\mathcal{O}((N \cdot P)^2)$, which do not participate in back propagation. To reduce such computational burden that undermines the application of the proposed method to large $N$, we train the model by partitioning the pre-defined spatial graph into multiple sub-graphs. In practice, we train REPST by sampling a sub-graph each time. By doing so, we can effectively reduce the computational costs and enable the model to scale to large $N$.

## 4 EXPERIMENTS

We thoroughly validate the effectiveness of REPST on various real-world datasets, including generative performance, overall forecasting performance and ablation study. We first introduce the experimental settings, including datasets and baselines. Then we conduct experiments to compare the few-shot, zero-shot and overall performance of our REPST with other previous works. Furthermore, we design comprehensive ablation studies to evaluate the impact of the essential components.

### 4.1 EXPERIMENTAL SETTINGS

**Datasets.** We conducted experiments on six commonly used real-world datasets (Song et al., 2020; Lai et al., 2018), each varying in the fields of traffic, solar energy, and air quality. The traffic datasets, Beijing Taxi (Zhang et al., 2017), NYC Bike [1], PEMS-BAY and METR-LA (Li et al., 2018), are collected from hundreds of individual detectors spanning the traffic systems across all major metropolitan areas of Beijing, NYC and California. The Air Quality[2] dataset includes six indicators (PM2.5, PM10, $NO_2$, CO, $O_3$, $SO_2$) to measure air quality, collected hourly from 35 stations across Beijing. Lastly, the Solar Energy dataset records variations every 10 minutes from 137 PV plants across Alabama, capturing the dynamic changes in solar energy production. Each dataset comprises tens of thousands of time steps and hundreds of nodes, offering a robust foundation for evaluating spatio-temporal forecasting models. The statistics of the datasets are summarized in Appendix A.2.

**Baselines.** We extensively compare our proposed REPST with the state-of-the-art forecasting approaches A.1, including (1) the GNN-based methods: Graph Wavenet (Wu et al., 2019), D2STGNN (Shao et al., 2022b) and MTGNN (Wu et al., 2020) (2) non-GNN-based state-of-the-art models: STID (Shao et al., 2022a), STAEformer (Liu et al., 2023a) and STNorm (Deng et al., 2021) which emphasizes the integration of spatial and temporal identities; (3) the state-of-the-art time series model: Informer (Zhou et al., 2021), iTransformer (Liu et al., 2023b) and PatchTST (Nie et al., 2022) (4) PLM-based time series forecasting models: FPT (Zhou et al., 2024); (5) methods with no trainable parameters: HI (Cui et al., 2021). We reproduce all of the baselines based on the original paper or official code.

### 4.2 REPST GENERALIZATION PERFORMANCE

**Few-Shot performance.** PLMs are trained using large amounts of data that cover various fields, equipping them with cross-domain knowledge. Therefore, PLMs can utilize specific spatio-temporal related textual representations to unlock their capabilities for spatio-temporal reasoning, which can

---

[1]https://github.com/LibCity/Bigscity-LibCity

[2]https://www.biendata.xyz/competition/kdd_2018/data/

Table 1: **Few-Shot** performance comparison on six real-world datasets in terms of MAE and RMSE. We utilize *data in one day* (less than 1%)for training and the same data as full training settings for validation and test. *The input history time steps $T$ and prediction steps $\tau$ are both set to 24.* We use the average prediction errors over all prediction steps. Bold denotes the best performance and underline denotes the second-best performance.

| Dataset | METR-LA | | PEMS-BAY | | Solar Energy | | Air Quality | | Beijing Taxi | | | | NYC Bike | | | |
|---|---|---|---|---|---|---|---|---|---|---|---|---|---|---|---|---|
| | | | | | | | | | Inflow | | Outflow | | Inflow | | Outflow | |
| Metric | MAE | RMSE | MAE | RMSE | MAE | RMSE | MAE | RMSE | MAE | RMSE | MAE | RMSE | MAE | RMSE | MAE | RMSE |
| Informer | 8.19 | 14.35 | 5.30 | 10.43 | 8.95 | 11.92 | 38.02 | 56.45 | 29.20 | 53.52 | 28.76 | 52.53 | 6.99 | 16.44 | 6.33 | 15.62 |
| iTransformer | 7.72 | 15.85 | 5.20 | 10.94 | 4.74 | 8.27 | 35.59 | 52.95 | 31.56 | 58.11 | 32.22 | 59.93 | 8.23 | 16.21 | 7.46 | 15.68 |
| PatchTST | 7.20 | 15.56 | 4.52 | 8.85 | 4.65 | 7.82 | 35.76 | 53.80 | 32.66 | 61.17 | 32.58 | 60.95 | 7.03 | 15.32 | 6.88 | 14.84 |
| MTGNN | 9.62 | 17.60 | 5.67 | 8.91 | 4.73 | 8.68 | 36.51 | 53.14 | 28.98 | 48.72 | 28.80 | 46.87 | 6.51 | 14.85 | 6.56 | 14.90 |
| GWNet | 7.04 | 12.58 | 5.84 | 9.42 | 9.10 | 11.87 | 36.26 | 54.88 | 29.24 | 51.68 | 29.47 | 50.52 | 12.55 | 21.97 | 12.68 | 22.27 |
| STNorm | 7.93 | 13.67 | 5.15 | 8.92 | 5.36 | 9.59 | 36.38 | 57.66 | 28.92 | 50.59 | 28.86 | 49.39 | 11.69 | 20.17 | 12.53 | 21.84 |
| D2STGNN | 6.41 | 11.57 | 5.31 | 9.39 | 8.80 | 11.26 | 40.77 | 55.07 | 36.73 | 58.70 | 36.06 | 66.01 | 10.64 | 18.96 | 10.33 | 18.43 |
| STID | 7.26 | 12.70 | 6.83 | 12.88 | 4.89 | 9.41 | 43.21 | 61.07 | 32.73 | 51.77 | 32.91 | 51.94 | 8.94 | 16.34 | 8.88 | 15.77 |
| STAEFormer | 6.35 | 11.38 | 5.37 | 9.35 | 4.66 | 12.57 | 37.68 | 53.39 | 28.88 | 49.86 | 28.06 | 48.13 | 12.50 | 20.77 | 11.84 | 20.88 |
| FPT | 6.80 | 11.36 | 4.55 | 9.71 | 10.59 | 13.92 | 36.62 | 51.33 | 41.66 | 74.87 | 43.28 | 77.84 | 12.97 | 20.06 | 12.72 | 20.11 |
| REPST | **5.63** | **9.67** | **3.61** | **7.15** | **3.65** | **6.74** | **33.57** | **47.30** | **26.85** | **45.88** | **26.30** | **43.76** | **5.29** | **12.11** | **5.66** | **12.85** |

handle the difficulties caused by data sparsity. To verify this, we further conduct experiments on each field to evaluate the predictive performance of our proposed REPST in data-sparse scenarios. Our evaluation results are listed in Table 1. Concretely, all models are trained on 1-day data from the train datasets and tested on the whole test dataset. REPST consistently outperforms other deep models and PLM-based time series forecasters in various datasets. This illustrates that our REPST can perform well on a new downstream dataset and is suitable for spatio-temporal forecasting tasks with the problem of data sparsity.

Specifically, our REPST show competitive performance over other baselines in few-shot experiments. It demonstrates that PLMs contain a wealth of spatio-temporal related knowledge from pre-training. Moreover, the capabilities of spatio-temporal reasoning can be enhanced by limited data. This shows a reliable performance when transferred to data-sparse scenarios.

**Zero-Shot performance.** In this part, we focus on evaluating the zero-shot generalization capabilities of our REPST within cross-domain and cross-region scenarios following the experiment setting of (Jin et al., 2023b). Specifically, we test the performance of a model on dataset A after training under a supervised learning framework on another dataset B, where dataset A and dataset B have no overlapped data samples. We use the similar experiment settings to full training experiments and evaluate on various cross-domain and cross-region datasets. The datasets includes NYC Bike, CHI Bike (Jiang et al., 2023), Solar Energy and Air Quality (NYC, CHI, Solar and Air). We compare our performance with recent works in time series or spatio-temporal data with open-sourced model weights (Das et al., 2023; Li et al., 2024c; Ekambaram et al., 2024).

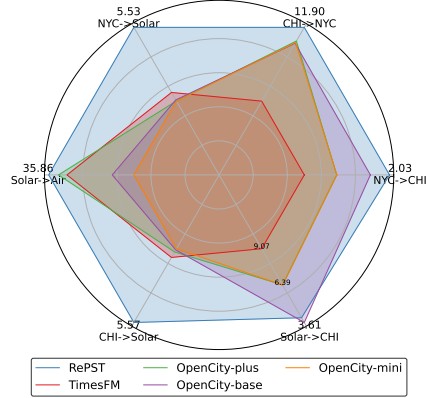

Figure 3: Zero-Shot Performance. We evaluate the zero-shot capability of our REPST in the same evaluation setting as few-shot experiments.

Our results in 3 show that REPST consistently secure top positions on all settings. This outstanding zero-shot prediction performance indicates REPST's versatility and adaptability in handling diverse scenarios. It does obtain transferable knowledge for dynamic systems by unlocking the reasoning capabilities of PLMs. Its excellent adaptation to brand new scenarios significantly reduces the time and computational resources typically required by traditional approaches. Although our REPST falls a little short to OpenCity-base in Solar Energy → CHI_Bike, it is because of the large amount of traffic related datasets included by OpenCity's pretrain datasets. Compared to it, our REPST is trained on Solar Energy dataset which has almost no connection with such traffic datasets. This relatively comparable performance demonstrate REPST's excellent generative capability in cross-domain settings.

Table 2: Performance comparison of **full training** on six real-world datasets in terms of MAE and RMSE. *The input history time steps $T$ and prediction steps $\tau$ are both set to 24.* We use the average prediction errors over all prediction steps. Bold denotes the best performance and underline denotes the second-best performance.

| Dataset | METR-LA | | PEMS-BAY | | Solar Energy | | Air Quality | | Beijing Taxi | | | | NYC Bike | | | |
| | | | | | | | | | Inflow | | Outflow | | Inflow | | Outflow | |
| Metric | MAE | RMSE | MAE | RMSE | MAE | RMSE | MAE | RMSE | MAE | RMSE | MAE | RMSE | MAE | RMSE | MAE | RMSE |
| --- | --- | --- | --- | --- | --- | --- | --- | --- | --- | --- | --- | --- | --- | --- | --- | --- |
| HI | 9.88 | 16.98 | 5.51 | 10.50 | 9.42 | 12.53 | 53.18 | 67.42 | 105.55 | 142.98 | 105.63 | 143.08 | 11.98 | 19.23 | 12.18 | 19.50 |
| Informer | 4.68 | 8.92 | 2.54 | 5.30 | 3.92 | 5.91 | 29.38 | 42.58 | 16.41 | 29.03 | 16.01 | 26.90 | 3.49 | 8.36 | 3.92 | 9.52 |
| iTransformer | 4.16 | 9.06 | 2.51 | 5.90 | 3.33 | 5.41 | 28.37 | 44.33 | 21.72 | 36.80 | 22.15 | 38.63 | 3.15 | 7.55 | 3.28 | 7.82 |
| PatchTST | 4.15 | 9.07 | 2.06 | 4.85 | 3.49 | 5.89 | 28.05 | 44.81 | 23.64 | 43.63 | 22.71 | 41.52 | 3.58 | 8.83 | 3.66 | 8.99 |
| MTGNN | 3.76 | 7.45 | 1.94 | 4.40 | 3.60 | 5.61 | 27.07 | 40.17 | 15.92 | 26.15 | 15.79 | 26.08 | 3.31 | 7.91 | 3.38 | 8.24 |
| GWNet | 3.93 | 8.19 | 2.28 | 5.06 | 3.55 | 5.39 | 31.57 | 44.82 | 15.69 | 26.82 | 15.76 | 26.84 | 3.13 | 7.58 | 3.33 | 7.64 |
| STNorm | 3.98 | 8.44 | 2.20 | 5.02 | 4.17 | 5.99 | 30.73 | 44.82 | 15.37 | 27.50 | 15.45 | 27.52 | 3.14 | 7.46 | 3.24 | 7.63 |
| D2STGNN | 3.94 | 7.68 | 2.11 | 4.83 | 4.36 | 5.85 | 27.77 | 41.87 | 24.33 | 45.65 | 26.86 | 45.57 | 3.10 | 7.43 | 3.25 | 7.75 |
| STID | 3.68 | 7.46 | 1.93 | 4.31 | 3.70 | 5.57 | 26.94 | 41.01 | 15.60 | 27.96 | 15.81 | 28.28 | 3.36 | 7.91 | 3.38 | 8.24 |
| STAEFormer | 3.60 | 7.44 | 1.97 | 4.33 | 3.44 | 5.21 | 28.12 | 41.83 | 15.47 | 26.45 | 16.08 | 26.83 | 3.03 | 7.39 | 3.27 | 7.56 |
| FPT | 6.03 | 10.85 | 2.56 | 5.01 | 6.02 | 8.31 | 32.79 | 47.55 | 32.41 | 55.28 | 32.77 | 55.77 | 7.21 | 12.76 | 7.75 | 13.85 |
| REPST | 3.63 | 7.43 | 1.92 | 4.33 | 3.27 | 5.12 | 26.20 | 39.37 | 15.13 | 25.44 | 15.75 | 25.24 | 3.01 | 7.33 | 3.16 | 7.43 |

**Increasing predicted length.** In this part, we analyze the model performance across varying prediction horizons $\tau \in \{6, 12, 24, 36, 48\}$, with a fixed input length T = 48. Figure 4 showcases the MAE and RMSE across two datasets: Air Quality and NYC Bike, for four models. The REPST model demonstrates the most stable performance across both MAE and RMSE metrics, particularly in longer prediction horizons ($\tau = 36, 48$). In contrast, previous state-of-the-art models exhibit notable performance degradation as the prediction horizon increases. The performance of REPST, on the other hand, remains relatively stable and robust, demonstrating its efficacy in leveraging PLM to improve performance over longer-term predictions, which can also be attributed to its ability to capture both spatial and temporal correlations effectively, making it highly suited for few-shot learning tasks in spatio-temporal forecasting.

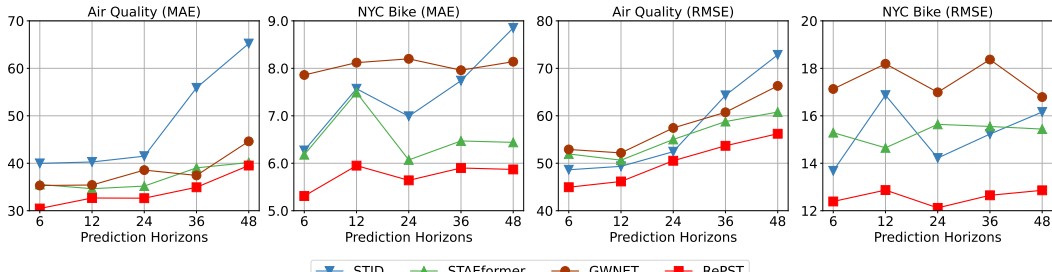

Figure 4: Few-Shot performance with multiple prediction horizons $\tau \in \{6, 12, 24, 36, 48\}$ and fixed input length $T = 48$. While the performance of previous state-of-the-art models keeps declining with the increasing of prediction length, the REPST framework empowers the pretrain knowledge of the reprogrammed spatial language model and obtains a relatively stable performance.

## 4.3 FULL TRAINING PERFORMANCE OF REPST

Table 2 reports the overall performance of our proposed REPST as well as baselines in 6 real-world datasets with the best in **bold** and the second underlined. As can be seen, REPST consistently achieves either the best or second-best results in terms of MAE and RMSE.

Notably, REPST surpasses the state-of-the-art PLM-based time series forecaster FPT (Zhou et al., 2024) by a large margin in spatio-temporal forecasting tasks, which can demonstrate that simply leveraging the PLMs cannot handle problems with complex spatial dependencies. Additionally, the performance of our REPST reaches either the best or second-best results in METR-LA and PEMS-BAY datasets. Previous state-of-the-art models, STAEformer and STID, learn global shared embeddings both in spatial structure and temporal patterns tailored for certain datasets, which is harmful to their generalization abilities but benefits their capabilities to handle single datasets. Our spatio-temporal reprogramming block leverages a wide range of vocabulary and sample words that

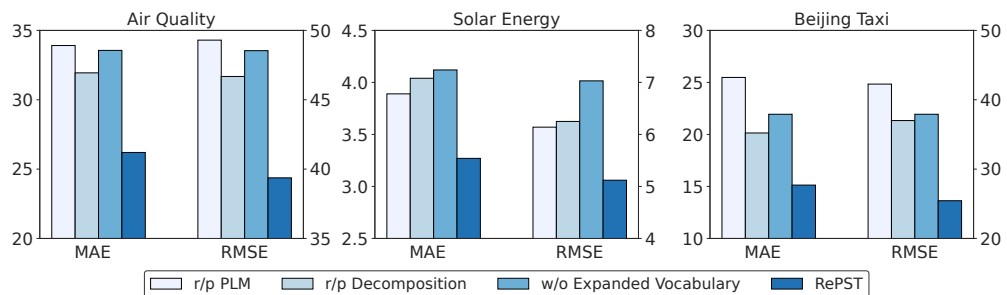

Figure 5: Ablation study. We conduct multiple detailed ablation studies on Air Quality, Solar Energy and Beijing Taxi datasets to figure out the effects of REPST's main components.

can adequately capture the spatio-temporal patterns, which do make an impact on unlocking the capabilities of PLMs to capture fine-grained spatio-temporal dynamics.

## 4.4 ABLATION STUDY

To figure out the effectiveness of each component in our REPST, we further conduct detailed ablation studies on Air Quality and Solar Energy datasets with three variants as follows:

- r/p PLM: it replaces the pre-trained language model backbone with transformer layers, following the setting of (Tan et al., 2024).
- r/p Decomposition: it replaces the physics-aware spatio-temporal decomposer with a transformer encoder.
- w/o expanded vocabulary: it removes our selective spatio-temporal vocabulary and utilizes the dense mapping function to enhance reprogramming.

Figure 5 shows the comparative performance of the variants above on Air Quality, Solar Energy and Beijing Taxi. Based on the results, we can make the conclusions as follows: (1) Our REPST actually leverage the pretrain knowledge and generative capabilities of PLMs. When we replace the PLM backbone with transformer layers, the performance of all the datasets decline obviously, indicating that the pretrain knowledge makes an effect to handle spatio-temporal dynamics. (2) The physics-aware spatio-temporal decomposer which adaptively disentangles input spatio-temporal data into physics-aware components can actually enable PLMs to better understand spatio-temporal dynamics. When constructing spatio-temporal dependencies by a transformer encoder layer, it is still unclear for PLMs to comprehend. (3) The impressive performance in w/o expanded vocabulary demonstrates that the selectively reconstructed spatio-temporal vocabulary achieves accurate reprogramming which enables PLMs to focus on relationships among tokens in 3D geometric space.

## 5 RELATED WORKS

### 5.1 SPATIO-TEMPORAL FORECASTING

Spatio-temporal forecasting has been playing a critical role in various smart city services, such as traffic flow prediction (Liu et al., 2023a; Shao et al., 2022a; Liu et al., 2020), air quality monitoring (Han et al., 2023; 2021), and energy management (Geng et al., 2019). Unlike traditional time series forecasting, the forecasting challenges associated with spatio-temporal data are often characterized by the unique properties of strong correlation and heterogeneity along the spatial dimension, which are inherently more complex.

Early studies usually capture spatial dependencies through a predefined graph structure (Li et al., 2018; Han et al., 2020; Shao et al., 2022b; Wu et al., 2020), which decribes the explicit relationships among different spatial locations. In recent years, there is a growing trend toward the utilization of adaptive spatio-temporal graph neural networks, which can automatically capture dynamic spatial graph structures from data. For instance, Graph WaveNet (Wu et al., 2019) eliminate the need for predefined graphs by learning an adaptive adjacency matrix using two embedding matrices. AGCRN (Bai et al., 2020) introduces a node adaptive parameter learning layer, allowing it to learn node-specific spatio-temporal patterns. Besides, attention mechanism is also widely employed in existing

models, as seen in examples like GMAN (Zheng et al., 2020), ASTGNN (Guo et al., 2021), and STAEformer (Liu et al., 2023a). In contrast, STID (Shao et al., 2022a), a model based on Multi-Layer Perceptrons (MLPs), achieves state-of-the-art results by utilizing multiple embedding techniques to memorize stable spatial and temporal patterns.

More recently, inspired by the huge success of PLMs in natural language processing field, there is increasing interest in building pre-trained models for spatio-temporal forecasting tasks. Several studies (Liu et al., 2024a;b; Yan et al., 2023; Jiang et al., 2024) explore the application of PLMs for handling spatio-temporal data. Among these, UrbanGPT (Li et al., 2024b) offers a promising end-to-end solution by integrating spatio-temporal data with textual information, enabling accurate predictions of urban dynamics. Furthermore, the strong power of Transformer offers an opportunity to build spatio-temporal foundation models, such as OpenCity (Li et al., 2024c) and UniST (Yuan et al., 2024). Trained on numerous spatio-temporal data, these models demonstrate strong capabilities and versatility across diverse forecasting scenarios. However, due to the problems of data-sparsity in multiple spatio-temporal scenarios, it is difficult for these models to gather large amount of data to perform pretraining comprehensively. In addition, the PLM-based spatio-temporal forecasting approaches do not fully leverage PLM's potential. To address these gaps, this paper introduces a new reprogramming framework to leverage PLM's generative and reasoning capabilities for spatio-temporal forecasting, particularly in data-sparse scenarios.

### 5.2 PRETRAINED LANGUAGE MODELS FOR TIME SERIES

In recent years, PLMs have demonstrated remarkable performance across various time series analysis tasks, including forecasting (Zhou et al., 2024; Gruver et al., 2024; Zhang et al., 2024), classification (Sun et al., 2023), and anomaly detection (Zhou et al., 2024). A significant body of research has focused on leveraging PLMs to address these challenges (Cao et al., 2023; Zhou et al., 2024; Gruver et al., 2023; Lai et al., 2023). However, a persistent issue in these efforts is the modality gap between time series data and natural language. To address this challenge, Time-LLM (Jin et al., 2023b) develops a time series reprogramming approach (Yang et al., 2021), which can effectively bridges the modality gap between time series and text data. The objective of reprogramming is to learn a trainable transformation function that can be applied to the patched time series data, enabling it to be mapped into the textual embedding space of the PLM.

Nevertheless, (Tan et al., 2024) conducted numerous experiments showing that existing PLM-based approaches (Zhou et al., 2024; Jin et al., 2023b; Wang et al., 2024) do not fully unlock the reasoning or generative capabilities of PLMs. The reason these approaches achieve high performance lies in the similar sequential formulation shared by time series and natural language (Liu et al., 2024d). In fact, even when PLMs are merely used as one-dimensional encoders, time series analysis tasks can still benefit from the pre-trained weights of PLMs. Although PLMs can handle one-dimensional sequential data like text and time series, they fall short in capturing dependencies among complex spatio-temporal structure, leading to suboptimal performance for spatio-temporal forecasting tasks. In this work, we propose REPST, which enables PLMs to comprehend complex spatio-temporal dynamics via a physics-aware decomposition-based reprogramming strategy.

## 6 CONCLUSION

In this paper, we highlight the underlying reason for the poor performance of previous PLM-based approaches in spatio-temporal forecasting, emphasizing the need for the interpretability to fully leverage PLM's potential. To address this problem, we developed REPST, a tailored spatio-temporal forecasting framework that enables PLMs to comprehend the complex spatio-temporal patterns via a physics-aware decomposition-based reprogramming strategy. We design a physics-aware spatio-temporal decomposer which adaptively disentangles complex spatio-temporal dynamics into components with rich physical semantics for PLM's better comprehension. Moreover, we construct an expanded spatio-temporal vocabulary by a selective approach, which enables PLMs to focus on relationships among 3D geometric space. As a result, PLM's potential is full unlocked to handle spatio-temporal forecasting tasks. Extensive experiments demonstrate that our proposed framework, REPST, achieves state-of-the-art performance on real-world datasets and exhibits exceptional capabilities in few-shot and zero-shot scenarios.

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

# A   IMPLEMENTATION DETAILS

## A.1   BASELINE MODELS

- **D2STGNN**: D2STGNN Shao et al. (2022b) is an advanced model designed to improve the accuracy and efficiency of traffic prediction by addressing the complexities inherent in spatial-temporal data. By decoupling spatial and temporal components, the model reduces complexity, making it more computationally efficient without sacrificing accuracy.

- **STAEformer**: STAEformer (Liu et al., 2023a) is a cutting-edge model that elevates the standard Transformer architecture for traffic forecasting by incorporating Spatio-Temporal Adaptive Embeddings. These embeddings dynamically encode both spatial and temporal dependencies, allowing the model to capture the complex, evolving patterns typical in traffic data. The spatial embeddings represent geographical relationships between traffic nodes, while the temporal embeddings account for time-related patterns like rush hours or seasonal variations. Unlike traditional static embeddings, it adaptively adjusts to the changing traffic conditions, enhancing the model's ability to predict future traffic flows with greater accuracy

- **Graph WaveNet**: Graph WaveNet (Wu et al., 2019) is a neural network model designed for spatio-temporal forecasting, particularly in graph-structured data like traffic networks. It combines graph convolutions to capture spatial dependencies between nodes (such as road intersections) and dilated temporal convolutions to model long-term and short-term trends over time. A key feature of Graph WaveNet is its learnable adjacency matrix, which dynamically adapts the relationships between nodes. It also uses diffusion convolutions to model the flow of information across the graph.

- **MTGNN**: MTGNN (Wu et al., 2020) is a model designed for forecasting tasks involving multivariate time series data with underlying graph structures, such as traffic or climate data. It combines graph neural networks to capture spatial dependencies between variables with temporal convolution layers to model time-based patterns. MTGNN uses an adaptive graph learning mechanism, where the graph structure representing relationships between variables is learned dynamically from the data, rather than being predefined. This allows the model to capture both static and dynamic dependencies in multivariate time series.

- **Informer**: Informer (Zhou et al., 2021) is a Transformer-based model specifically designed for long-range time-series forecasting, addressing the computational challenges of handling large sequences. It introduces the ProbSparse self-attention mechanism, which selectively focuses on the most important queries, reducing the computational load while maintaining accuracy. Additionally, Informer incorporates a distilling mechanism, which progressively reduces the length of the time series, retaining only essential features and improving efficiency. These innovations allow Informer to handle large-scale time series data efficiently, making it particularly effective for long-range forecasting tasks such as weather prediction, traffic flow analysis, and energy consumption forecasting.

- **iTransformer**: iTransformer (Liu et al., 2023b) is an advanced model designed for multivariate time series data forecasting, leveraging the strengths of the Transformer architecture to effectively capture spatial dependencies between variables. Unlike traditional Transformers, iTransformer employs a novel invert mechanism that adapts to the unique characteristics of spatio-temporal data, enabling it to focus on relevant spatial features while modeling long-range temporal patterns. By effectively integrating cross-variables information, iTransformer achieves high accuracy and efficiency in forecasting tasks that involve complex, interrelated data.

- **PatchTST**: PatchTST (Nie et al., 2022) is a novel framework tailored for time series forecasting that utilizes a patch-based approach to capture temporal dynamics efficiently. By dividing the input data into patches and employing a transformer architecture, it enhances the model's ability to learn local and global patterns simultaneously. This design not only improves predictive performance but also reduces computational complexity, making it suitable for large-scale time series applications across various domains, including finance, healthcare, and IoT.

- **STNorm**: STNorm Deng et al. (2021) normalizes data to better capture underlying patterns in both spatial and temporal dimensions. By addressing the variability in data across different

time steps and locations, STNorm improves the accuracy of predictions, offering a robust approach to handling complex, dynamic datasets.

- **STID**: STID Shao et al. (2022a) emphasizes the integration of spatial and temporal identities to enhance predictive performance. It employs unique identifiers for spatial and temporal components to effectively capture and utilize the inherent structure and patterns in the data.

- **FPT**: FPT Zhou et al. (2024) demonstrate that partly frozen pre-trained models on natural language or images can handle all main time series analysis tasks.

- **HI**: HI Cui et al. (2021) is a baseline model designed to leverage the natural continuity of historical data without relying on trainable parameters. HI directly uses the historical data point closest to the prediction target within the input time series as the forecasted value. HI capitalizes on the inherent persistence of historical patterns, making it a simple yet effective benchmark for time series forecasting tasks.

## A.2 DATASET DESCRIPTIONS

We follow the same data processing and train-validation-test set split protocol used in the baseline models, where the train, validation, and test datasets are strictly divided according to chronological order to make sure there are no data leakage issues. As for the forecasting settings, we fix the length of the lookback series as $24$, and the prediction length is $24$. Six commonly used real-world datasets vary in fields of traffic (PEMS-BAY, METR-LA, Beijing Taxi, NYC Bike), energy (Solar Energy) and air quality (Air Quality), each of which holds tens of thousands of time steps and hundreds of nodes. Beijing Taxi and NYC Bike datasets are collected in every 30 minutes from tens of individual detectors spanning the traffic system across all major metropolitan areas of NYC and Beijing, which are widely used in previous spatio-temporal forecasting studies. PEMS-BAY and METR-LA datasets are collected in every 5 minutes from nearly 40,000 individual detectors spanning the highway system across all major metropolitan areas of California. Air Quality dataset holds 6 indicators ($PM2.5$, $PM10$, $NO_2$, $CO$, $O_3$, $SO_2$) to measure air quality. They are collected from 35 stations in every 1 hour. And Solar Energy dataset collect the every 10 minutes variations of 137 PV plants across Alabama. Notably, we construct the graph for 35 stations by leveraging series similarity between nodes. The details of datasets are provided in Table 3

Table 3: Detailed dataset descriptions. *Dim* denotes the variate number of each dataset. *Dataset Participation* denotes the total number of time points in (Train, Validation, Test) split respectively. *Frequency* denotes the sampling interval of time points.

| Dataset | Dim | Dataset Participation | Frequency | Information |
|---|---|---|---|---|
| Air Quality | 35 | (6075, 867, 1736) | 1 hour | Air Quality |
| PEMS-BAY | 325 | (35488, 5207, 10414) | 5 min | Traffic Speed |
| METR-LA | 207 | (23958, 3422, 6845) | 5 min | Traffic Speed |
| Beijing Taxi(Inflow) | 1024 | (3831, 547, 1095) | 30 min | Taxi Service |
| Beijing Taxi(Outflow) | 1024 | (3831, 547, 1095) | 30 min | Taxi Service |
| NYC Bike(Inflow) | 128 | (3058, 437, 874) | 30 min | Bike Service |
| NYC Bike(Outflow) | 128 | (3058, 437, 874) | 30 min | Bike Service |
| CHI Bike | 270 | (6183, 883,1766) | 30 min | Bike Service |
| Solar Energy | 137 | (36776, 5254, 10507) | 10 min | Energy |

## A.3 EVALUATION METRICS

Three metrics are used for evaluating the models: mean absolute error (MAE) and root mean squared error (RMSE). Lower values of metrics stand for better performance. RMSE and MAE measure

absolute errors, while MAPE measures relative errors.

$$\text{MAE} = \frac{1}{N} \sum_{i=1}^{N} \left| \hat{\mathbf{y}}^i - \mathbf{y}^i \right|,$$

$$\text{RMSE} = \sqrt{\frac{1}{N} \sum_{i=1}^{N} (\hat{\mathbf{y}}^i - \mathbf{y}^i)^2},$$

where $\hat{\mathbf{y}}^i, \mathbf{y}^i$ represents a sample from $\hat{\mathbf{Y}}$ and $\mathbf{Y}$, and N represent the total number of samples.

### A.4    KOOPMAN THEORY

Koopman Theory (Koopman, 1931) shows that any nonlinear dynamic system, including spatio-temporal series, can be modeled by an infinite-dimensional linear Koopman operator acting on a space of measurement functions.

Koopman operator theory provides a powerful framework for analyzing nonlinear dynamical systems by lifting them into a linear infinite-dimensional space. The Koopman framework has shown particular utility in analyzing spatio-temporal dynamic systems, where complex, nonlinear behaviors can be represented using linear superpositions of Koopman eigenfunctions. The Koopman operator, denoted as $\mathcal{K}$, is a linear operator that acts on observable functions of the system state, rather than directly on the state space itself. In a nonlinear dynamical system described by $\mathbf{x}_{t+1} = \mathbf{f}(\mathbf{x}_t)$, where $\mathbf{x} \in \mathbb{R}^n$ is the state and $\mathbf{f}$ is a nonlinear map, the Koopman operator is defined as:

$$\mathcal{K}g(\mathbf{x}_t) = g(\mathbf{f}(\mathbf{x}_t)), \tag{9}$$

where $g$ is an observable, a scalar-valued function that maps the system's state to a measurable quantity. The key insight is that while the system dynamics may be nonlinear, the evolution of observables under the action of the Koopman operator is linear. This allows for the application of spectral analysis techniques to extract meaningful modes of the system's dynamics. Central to Koopman analysis are the Koopman eigenfunctions, $\phi(\mathbf{x})$, which satisfy:

$$\mathcal{K}\phi(\mathbf{x}) = \lambda\phi(\mathbf{x}), \tag{10}$$

where $\lambda$ is the associated Koopman eigenvalue. The eigenfunctions provide a coordinate system in which the dynamics of the system are fully described by linear evolution:

$$\phi(\mathbf{x}_{t+1}) = e^{\lambda t}\phi(\mathbf{x}_t). \tag{11}$$

In practice, the Koopman spectrum, which consists of the eigenvalues $\lambda$, determines the growth, decay, or oscillatory behavior of different dynamic modes within the system. This makes it an invaluable tool for decomposing complex, high-dimensional spatio-temporal dynamics into simpler, interpretable components.

### A.5    EVOLUTIONARY DECOMPOSITION

Decomposition methods based on the eigenvectors of a dynamic system's evolution matrix, such as Dynamic Mode Decomposition (Schmid, 2010; Kutz et al., 2016), often have more explicit physical interpretations compared to Fourier-based methods. It captures both transient (non-periodic) and periodic dynamics, as the eigenvalues can describe exponentially growing or decaying modes. In is derived from data representing the true dynamics of the system, which can separate modes that correspond to specific physical processes, such as fluid flow patterns, mechanical oscillations, or heat transfer (Proctor et al., 2016; Brunton et al., 2016; Chen et al., 2012). A dynamic system can be represented in a state-space form as:

$$\frac{dx(t)}{dt} = \mathcal{A}x(t) \tag{12}$$

where $x(t)$ is the state vector at time $t$, $\mathcal{A}$ is the evolutionary matrix that describes the dynamics of the system. The solution to this differential equation can be expressed using the matrix exponential:

$$x(t) = e^{\mathcal{A}t}x(0), \tag{13}$$

where $e^{\mathcal{A}t}$ is the matrix exponential of $\mathcal{A}$ and represents the evolution of the state over time.

Specifically, we can approximate the evolutionary matrix $\mathcal{A}t$ in the following steps. Given an observation $\mathbf{X}_{1:m}$ representing the dynamic system's state at discrete time intervals, we organize the data into two matrices:

$$\mathbf{X}_{1:t-1} = [\mathbf{x}_1, \mathbf{x}_2, \ldots, \mathbf{x}_{t-1}] \in \mathbb{R}^{n \times (t-1)} \tag{14}$$

$$\mathbf{X}_{2:t} = [\mathbf{x}_2, \mathbf{x}_3, \ldots, \mathbf{x}_t] \in \mathbb{R}^{n \times (t-1)}, \tag{15}$$

Here, $\mathbf{x}_i \in \mathbb{R}^n$ represents the state of the system at the $i$-th time step, while $t$ denotes the number of snapshots. The evolutionary matrix $\mathcal{A}t$ maps the two data matrices such that:

$$\mathbf{X}_{2:t} \approx \mathcal{A}\mathbf{X}_{1:t-1}, \tag{16}$$

Then, we get the mathematical expression of $\mathcal{A}$, formulated as:

$$\mathcal{A} \approx \mathbf{X}_{2:t}\mathbf{X}_{1:t-1}^+, \tag{17}$$

where $X_{1:t-1}^+$ is the pseudoinverse of $X_{1:t-1}$. Assuming $\mathcal{A}$ is diagonalizable, we can express the evolution of the system using the matrix exponential. The eigenvalue decomposition of $\mathcal{A}$ is given by:

$$\mathcal{A} = VDV^{-1}, \tag{18}$$

where $D = \text{diag}(\omega_1, \omega_2, \ldots, \omega_n)$ is a diagonal matrix of eigenvalues $\omega_i$, $V = [v_1, v_2, \ldots, v_n]$ is the matrix of corresponding eigenvectors. We can further write:

$$\mathcal{A}v_i = \omega_i v_i, \tag{19}$$

The matrix exponential can be computed using the Jordan canonical form or the spectral decomposition:

$$e^{\mathcal{A}t} = Ve^{Dt}V^{-1}, \tag{20}$$

and the matrix exponential of D is computed as:

$$e^{Dt} = \text{diag}(e^{\omega_1 t}, e^{\omega_2 t}, \ldots, e^{\omega_n t}), \tag{21}$$

So we get the mathematical expression of the state vector at any time $t$:

$$\mathbf{x}_t \approx V\text{diag}(e^{\omega_1 t}, e^{\omega_2 t}, \ldots, e^{\omega_n t})V^{-1}\mathbf{x}_0, \tag{22}$$

$$\mathbf{x}_t \approx \sum_{i=1}^{C} \varepsilon_i e^{\omega_i t} v_i, \tag{23}$$

where $C$ represents the number of eigenvalues and $\varepsilon_i$ is calculated from $\mathbf{x}_0 = \sum_{i=1}^{C} \varepsilon_i v_i$. Specifically, we get the physics-aware dynamic components $\mathbf{X}_{dyn}$:

$$\mathbf{X}_{dyn} = \|_{i=0}^{C} \varepsilon_i e^{\omega_i t} v_i, \tag{24}$$

$$\mathbf{X}_{rec} = \sum_{i=0}^{C} \varepsilon_i e^{\omega_i t} v_i \tag{25}$$

We further introduce the evolutionary reconstruction of dynamic system based on analysis of eigenvalues $\omega_i$, $V = [v_1, v_2, \ldots, v_n]$. To further determine whether a mode is dominant, we can analyze its energy contribution (Schmid, 2010; Proctor et al., 2016; Kutz et al., 2016) to the overall system. The energy contribution of a mode is typically calculated using the following formula:

$$E_i = |\omega_i|^2 \cdot e^{2\text{Re}(v_i)t} \tag{26}$$

This formula combines the initial amplitude of the mode $\omega_i$ with the real part of the eigenvalue $\text{Re}(v_i)$, providing an estimate of the mode's energy contribution at time $t$ . If the energy contribution of a particular mode is significantly higher than that of other modes, it can be considered a dominant mode. Then we sort the values of $E_i$ for each mode $\omega_i$, and formulate the top-k mode $\omega_i$ as our selected most dominant modes $\alpha$.

# B EXPERIMENTAL DETAILS

## B.1 HYPER PARAMETER SETTINGS

For our prediction tasks, we aim to predict the next 24 steps of data based on the previous 24 steps. Both the historical length ($T$) and prediction length ($\tau$) are set to 24. Moreover, the parameters for the convolution kernel in patch embedding layers are set to 3 and the number of the multi-head attention larers of reprogramming layer is set to 1. Additionally, we obtain the embedding of patches with the dimension of 64.

## B.2 FURTHER EXPERIMENTAL SETUP DESCRIPTIONS

During the reprogramming phrase, we sample 1000 most relevant words to capture the complex dynamic spatio-temporal dependencies. It is important to note that the missing data of the training dataset are filled by the former time step of the same node. By doing so, it helps to improve the performance of pre-trained language models to handle spatio-temporal time series tasks. Because the value 0 could disturb the capabilities of pre-trained language models for understanding the consistent textual series. Similar to traditional experimental settings, each time series is split into three parts: training data, validation data, and test data. For the few-shot forecasting task, only a certain percentage timesteps of training data are used, and the other two parts remain unchanged. The evaluation metrics remain the same as for classic spatio-temporal time series forecasting. We repeat this experiment 3 times and report the average metrics in the following experiments. Additionally, before the training procedure, the Fourier representations of each node are pre-calculated to save reduce the computation cost, as a result of which to reduce the training time cost. All the experiments are implemented in PyTorch and conducted on a single NVIDIA RTX 3090 24GB GPU. We utilize ADAM with an initial learning rate of 0.002 and MAE loss for the model optimization. We set the number of frozen GPT-2 blocks in our proposed model $gpt\_layers \in (3, 6, 9, 12)$. The dimension of patched representations $D$ is set from $\{64, 128, 256\}$. All the compared baseline models that we reproduced are implemented based on the benchmark of BasicTS Shao et al. (2023), which is developed based on EasyTorch, an easy-to-use and powerful open-source neural network training framework.

| Category | REPST | TimesFM | OC-plus | OC-base | OC-mini | FPT | Time-LLM |
|----------|-------|---------|---------|---------|---------|------|----------|
| NYC → CHI | 2.03 | 9.07 | 6.39 | 3.61 | 6.38 | 12.56 | 10.32 |
| CHI → NYC | 11.9 | 19.23 | 13.26 | 13.48 | 13.4 | 30.24 | 25.44 |
| NYC → Solar | 5.53 | 9.81 | 10.37 | 10.3 | 10.38 | 22.36 | 18.04 |
| Solar → Air | 31.86 | 38.62 | 37.34 | 45.44 | 48.71 | 68.44 | OOT |
| CHI → Solar | 5.57 | 9.81 | 10.17 | 10.3 | 10.38 | 26.32 | 16.28 |
| Solar → CHI | 3.96 | 9.07 | 6.39 | 3.61 | 6.38 | 15.44 | OOT |

Table 4: Zero-Shot performance comparison of different models across various dataset transfers. OOT indicates Out-Of-Time errors for specific tasks.

## B.3 DETAILED COMPARISON OF ZERO SHOT PERFORMANCE

We provide the complete numerical results corresponding to Figure 3 in Table 4.

Regarding the limited number of models compared for zero-shot performance, this is primarily due to the current scarcity of open-source spatio-temporal forecasting models explicitly designed for zero-shot capabilities. To the best of our knowledge, the baselines we included represent the available state-of-the-art in this area. However, we acknowledge the importance of broader comparisons, so we conducted more zero-shot experiments on time series models (FPT, Time-LLM). While these models may not be specifically designed for spatio-temporal zero-shot forecasting tasks, their inclusion will provide a more comprehensive context for evaluation.

As can be seen in the table, both FPT and Time-LLM fall short in such spatio-temporal prediction tasks, primarily due to the lack of abilities for spatial modeling. Moreover, because of the huge computational cost of Time-LLM caused by large amount of spatial variables, we could only complete experiments on relatively small datasets (OOT for out of time). This further demonstrates the superiority of our model in handling spatio-temporal tasks compared to the time series models.

## C    SHOW CASES

To provide the visualization of the prediction effect, we list the prediction showcases of certain nodes contained in dataset PEMS-BAY. Concretely, we visualize the input observation and prediction in 24 steps of four nodes from the node set (Figure 6).

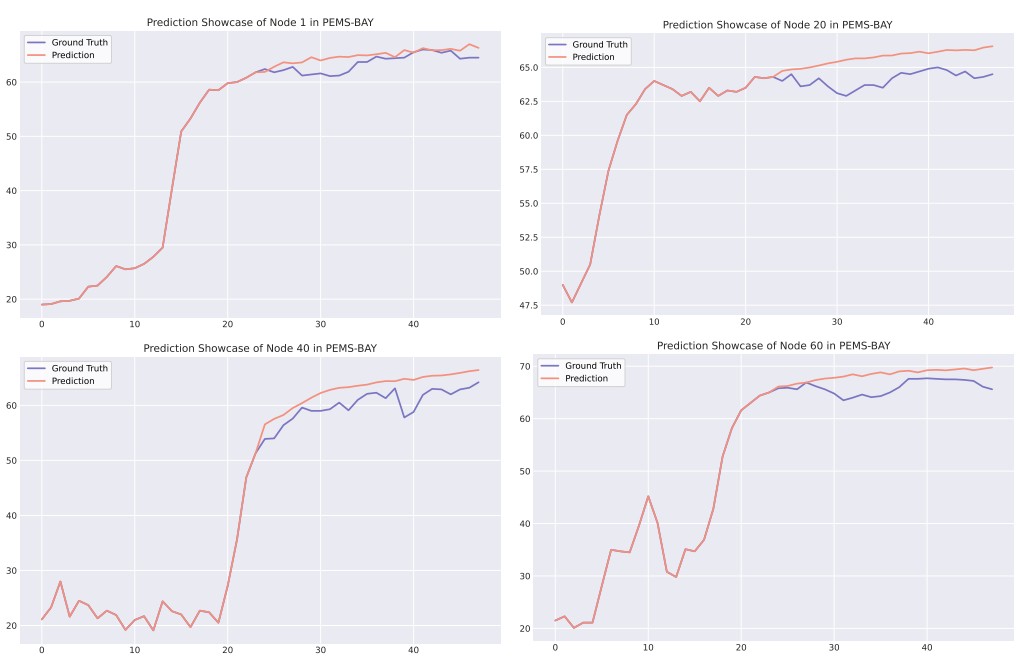

Figure 6: Case Study for PEMS-BAY. We show the input observation in 24 time steps and prediction horizon as 24.By showing input, ground truth and prediction together, we can get a clear understanding to the model performance.

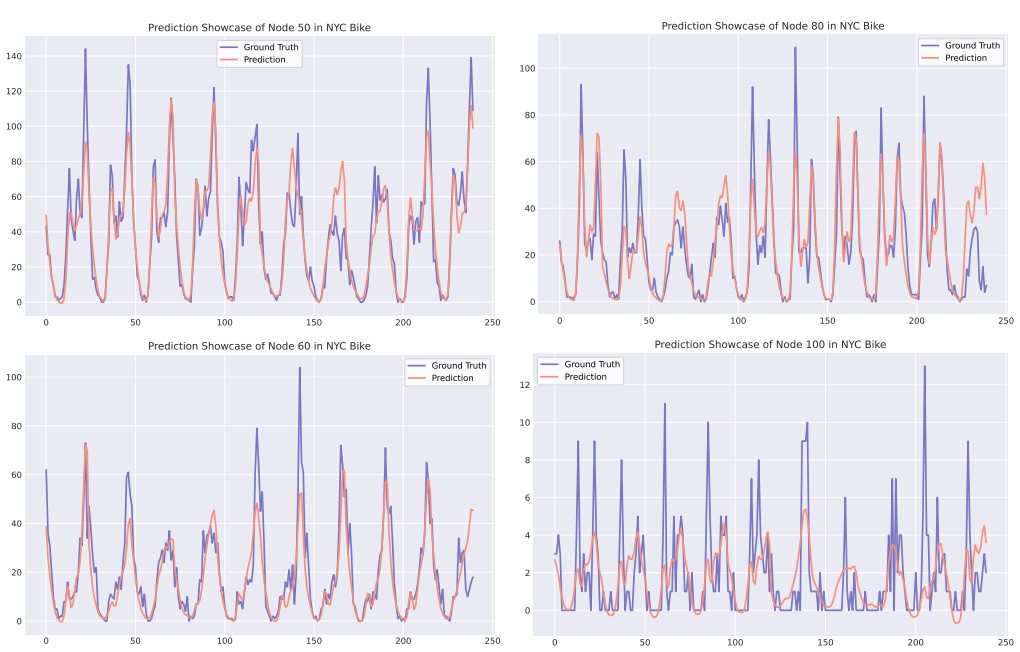

Figure 7: Case Study for NYC Bike. We show the ground truth and predictions for 240 time steps in one figure to grasp a global perception.

We also conduct cases from NYC Bike, which show the predictions in 240 time steps from certain nodes to grasp a global perception.

**Selected Vocabulary**

Annual, Begin, rival, Bloom, refreshed, Gained, Wide, Split, incre, imitation, extremely, rising, few, inserted, core, air, ight, slowdown, cause, dawn, soon, erie, rapid, recalls, Develop, wer, St, Less, industry, move, icted, bad, regained, rain, atmosp, wind, unchanged, dusty, , itialized, intervals, Benefit, crease, spread, holding, ...

Figure 8: Visualization of selected spatio-temporal vocabulary.

We further show cases of our expanded spatio-temporal vocabulary by visualizing the selective words (see Figure 8). Noted that these selective word embeddings are encodings for word or word morphemes, which can be regarded as the smallest unit that makes up a word. To intuitive display, we artificially combine these small units into words. In our RePST, this part is completed by cross attention module, which can automatically match the spatio-temporal data with most relevant words. Specifically, in order to minimize the subjective impact, we decrease the number of our expanded spatio-temporal vocabulary as 100. Finally, we present a case study on real-world spatio-temporal datasets. We sample the lookback window of one single node in Air Quality and visualize the selected vocabulary learned by differentiable discrete reprogramming blocks. As shown in tables below, it can be clearly observed that words from the selected vocabulary can jointly describe the temporal trend along with the spatio-temporal pattern vividly, indicating the effectiveness of reprogramming spatio-temporal data into textual representations.

We also observed the insightful interpretability of our framework. For temporal components, series trends are reflected with words like "increase" Words like "Annual", "rapid", and "unchanged" also show temporal patterns. For spatial components, we get words for relative relations such as "move" and "spread". Furthermore, some words describe the pattern in certain scenarios vividly. "dusty", "rain" and "wind" represent a kind of phenomena which have strong relationship with air quality. The above results demonstrate that RePST effectively captures the characteristics of different scenarios and can be used for various downstream tasks.

## D    DISCUSSION

### D.1    DISCUSSIONS ON PHYSICAL KNOWLEDGE FROM DECOMPOSER

There are maybe questions that "Is the information extracted from spatio-temporal decomposer truly physical knowledge or fine-grained feature?". To answer this question, we claim as follow:

**Physics-Aware Justification.** Although DMD is designed to extract dynamic modes directly from observational data, numerous studies have demonstrated its capability to identify propagating waves, oscillatory behaviors, and decay patterns, which are strongly associated with dominant physical phenomena (Rowley et al., 2009; Tu, 2013; Yu et al., 2024). For example, research has shown that DMD can reveal vortex shedding patterns and periodic oscillations in fluid dynamics (Rowley et al., 2009). These patterns are closely aligned with the Navier-Stokes equations, reflecting the intrinsic physical dynamics of fluid motion (Tu, 2013; Yu et al., 2024). For this reason, DMD is particularly suitable for capturing physics-aware patterns from spatio-temporal data, such as traffic flow and air pollution, which exhibit similar regularities with those governing fluid dynamics. For instance, STDEN (Ji et al., 2022) and AirPhyNet (Hettige et al., 2024) model the dynamics of traffic flow and air pollution as a continuous diffusion process via differential equations inspired by fluid dynamics modeling.

**Why DMD Over PCA or Eigenvectors?** Unlike DMD, PCA or eigenvector-based methods are purely statistical tools that do not inherently account for the underlying physical behavior of a system. By analyzing the dominant modes, DMD can uncover patterns that are both interpretable and consistent with the underlying physical mechanisms, providing valuable insights for PLM that can improve predictive accuracy.

To validate our choice, we first conducted experiments by replacing the physical decomposer with PCA (see Table 5). This results in a noticeable decline in model performance, suggesting that PCA's inability to benefit PLM in our context. Furthermore, when we randomly initialized the pretrained weights of the PLM, the impact on performance was minor, further indicating that PCA could not effectively leverage the pretrained knowledge. These findings underscore the superiority of DMD in generating physically consistent interpretations and unlocking pretrained model knowledge.

Furthermore, to verify the effectiveness of the knowledge extracted by the physical decomposer, we conducted experiments that apply the decomposed data to two state-of-the-art spatio-temporal forecasting models, i.e., STID and GWNet (see Table 5). The results showed only marginal improvement, indicating that even advanced spatio-temporal forecasting models struggle to effectively utilize the physical knowledge. This demonstrates that the knowledge extracted by the physical decomposer is not simply a set of fine-grained features but represents a unique form of information that cannot be easily leveraged by all models.

In summary, DMD demonstrates a clear advantage over simpler methods such as PCA or eigenvector decomposition by integrating spatial and temporal dynamics through interpretable modes aligned with physical phenomena. Our additional experimental results further emphasize its ability to enhance PLM's performance by capturing meaningful, physics-aware patterns rather than merely extracting fine-grained statistical features.

Table 5: Performance comparison of **few shot** on real-world datasets in terms of MAE. PCA_REPST: REPST with PCA as decomposer; PCA_random_REPST: REPST with PCA as decomposer and randomly initialize the weights of the PLM; GWNet_decomposer: GWNet using decomposed features; STID_decomposer: STID with decomposed features; REPST_random: randomly initialize the weights of the PLM in REPST.

| Dataset | Air Quality | | Solar Energy | | NYC Bike | | | |
|---|---|---|---|---|---|---|---|---|
| | | | | | Inflow | | Outflow | |
| Metric | MAE | RMSE | MAE | RMSE | MAE | RMSE | MAE | RMSE |
| PCA_REPST | 37.49 | 51.66 | 4.61 | 9.74 | 7.51 | 15.33 | 7.81 | 15.16 |
| PCA_random_REPST | 38.04 | 54.32 | 4.67 | 10.84 | 7.55 | 15.78 | 7.82 | 15.23 |
| GWNet_decomposer | 35.81 | 51.94 | 8.94 | 11.02 | 11.84 | 20.67 | 10.96 | 19.48 |
| GWNet | 36.26 | 54.88 | 9.10 | 11.87 | 12.55 | 21.97 | 12.68 | 22.27 |
| STID_decomposer | 42.44 | 59.68 | 4.34 | 8.75 | 8.07 | 16.04 | 8.04 | 16.55 |
| STID | 43.21 | 61.07 | 4.89 | 9.41 | 8.94 | 16.34 | 8.88 | 15.77 |
| REPST_random | 40.12 | 56.27 | 5.21 | 9.32 | 7.83 | 16.41 | 6.81 | 15.82 |
| REPST | 33.57 | 47.30 | 3.65 | 6.74 | 5.29 | 12.11 | 5.66 | 12.85 |

D.2 DISCUSSIONS ON REASONING ABILITY OF PLMS

First, in our work, reasoning refers to the ability to make predictions by comprehending both spatial and temporal contexts. While zero-shot performance improvements demonstrate generalization, we argue that they also reflect enhanced reasoning capabilities. In zero-shot setting, the model encounters spatial regions or domains it has not seen during training. The model's predictive performance solely relies on its inherent reasoning capabilities and its capacity to infer patterns from prior spatio-temporal knowledge.

Second, our DMD-based decomposer inherently captures spatial correlations within spatio-temporal data by leveraging the co-evolving nature of spatial nodes. Specifically, the input matrix X, where rows represent spatial nodes and columns represent time steps, implicitly embeds spatial relationships through the correlated dynamics of the nodes. During decomposition, spatial nodes exhibiting similar temporal behaviors (e.g., traffic flow on adjacent roads) are naturally grouped into the same dynamic

mode, i.e., dominant global patterns that summarize the behavior of correlated spatial nodes. These patterns reflect how the spatio-temporal system as a whole evolves over time. Furthermore, DMD does not require prior spatial knowledge (e.g., spatial adjacency matrices), making it a flexible approach for datasets with implicit spatial relationships.

Third, we further conducted empirical studies to validate the contribution of DMD to spatial reasoning. In specific, we decomposed each node in the dataset independently and then concatenated them together, rather than decomposing the entire spatio-temporal system. By doing so, we can eliminate the impact of DMD on spatial dimension. We observe an obvious decrease in the model performance, indicating that the decomposer extract spatio-temporal knowledge from the system rather than simple temporal embeddings.

Table 6: Performance comparison of **few shot** on real-world datasets in terms of MAE and RMSE. *The input history time steps $T$ and prediction steps $\tau$ are both set to 96.*

| Dataset | Air Quality | | Solar Energy | | NYC Bike | | | |
| --- | --- | --- | --- | --- | --- | --- | --- | --- |
| | | | | | Inflow | | Outflow | |
| Metric | MAE | RMSE | MAE | RMSE | MAE | RMSE | MAE | RMSE |
| iTransformer | 50.92 | 72.95 | 4.58 | 9.91 | 4.45 | 10.34 | 4.45 | 10.33 |
| PatchTST | 44.22 | 66.67 | 4.24 | 8.56 | 4.47 | 10.88 | 4.48 | 10.88 |
| STID | 38.92 | 58.91 | 4.61 | 8.93 | 4.46 | 10.38 | 4.43 | 10.32 |
| GWNet | 40.47 | 60.44 | OOM | OOM | 7.59 | 16.43 | 7.53 | 16.28 |
| REPST | 34.34 | 54.38 | 4.02 | 8.05 | 4.40 | 9.95 | 4.39 | 9.87 |

### D.3 DISCUSSIONS ON LONGER PREDICTION HORIZONS

As is common in spatio-temporal forecasting research, our experiments focused on relatively short prediction lengths to align with standard evaluation protocols in the field. However, we acknowledge the importance of exploring longer prediction horizons to provide a more comprehensive assessment of model performance. We conducted additional experiments with a prediction length of 96 across four datasets (OOM: Out-Of-Memory).

As can be seen in Table 6, REPST consistently outperformed powerful baseline models even in long-term predictions, undersoring the robustness of our approach across varying prediction lengths.

### D.4 DISCUSSIONS ON THE EFFECTIVENESS OF RECONSTRUCTION

Reconstruction plays a pivotal role in DMD-related techniques, ensuring that the extracted modes faithfully capture the system's underlying dynamics while eliminating redundant noise. In our approach, we enhance this process by prioritizing the most dominant modes, enabling a more precise representation of key spatio-temporal patterns. We conducted experiments (Figure 9) to systematically evaluate its role and assess its influence on the overall results, following the setting of the ablation study in the paper.

As can be seen in the table above, the model performance declines obviously when we remove either dynamic components ($\mathbf{X}_{dyn}$) or reconstruction data ($\mathbf{X}_{rec}$). The ablation studies confirm that the reconstruction matrix is essential for maintaining high performance, alongside the dynamics component. The results also demonstrate the necessity of integrating both components to fully realize the advantages of the REPST framework.

## E BROADER IMPACT

### E.1 IMPACT ON REAL-WORLD APPLICATIONS

Our work copes with real-world spatio-temporal forecasting, which is faced with problems of data sparsity and intrinsic non-stationarity that poses challenges for deep models to train a domain foundation model. Since previous works thoroughly explore the solutions to deal with various spatio-temporal dependencies, we propose a novel approach which leverage the power of pre-trained

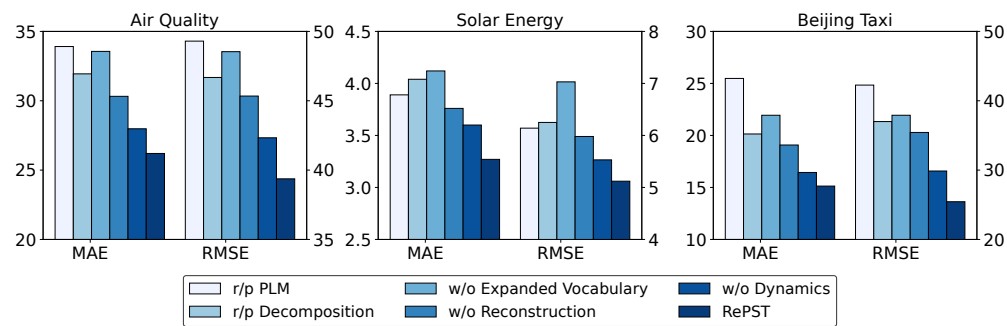

Figure 9: Ablation study. We conduct multiple detailed ablation studies on Air Quality, Solar Energy and Beijing Taxi datasets to figure out the effects of REPST's main components.

language models to handle spatio-temporal forecasting tasks, which fundamentally considers the natural connection between spatio-temporal information and natural language and achieves modality alignment by leveraging reprogramming. Without additional effort on prompts engineering(Li et al., 2024b; Yan et al., 2023) which is a time-cost but essential part in enhancing the capabilities of pre-trained language models, our REPST automatically learns the spatio-temporal related vocabulary which can unlock the domain knowledge of pre-trained language models to do spatio-temporal reasoning and predictive generation. Our model reaches state-of-the-art performance on the four real-world datasets , covering energy, air quality and transportation, and demonstrates remarkable capability to handle problem of data sparsity. Therefore, the proposed model makes it promising to tackle real-world forecasting applications, which can help our society to prevent multiple risks in advance with limited computational cost and small amount of data.

### E.2 IMPACT ON FUTURE RESEARCH

In this paper, we find that models trained on natural languages can handle spatio-temporal forecasting tasks, which is totally a different data modality from natural language. This demonstrates that aligning different data modality properly can unlock the domain knowledge obtained by the pre-trained model during the training process. Therefore, there is a possibility that models that pre-trained on data from various domains hold the capability to handle problems in different fields even if in different modalities. The underlying reasons why pre-trained models can handle cross-modality tasks still remains to be explain.

