# OpenReview forum: "Language Model Empowered Spatio-Temporal Forecasting via Physics-Aware Reprogramming"
_ICLR.cc/2025/Conference — Submitted to ICLR 2025_

### Official Review · Reviewer_EE3c · 2024-10-28

**Soundness:** 3
**Presentation:** 3
**Contribution:** 2
**Rating:** 6
**Confidence:** 3

**Summary:**

This paper presents a physics-aware PLM reprogramming framework REPST tailored for spatio-temporal forecasting. The proposed REPST consists of a physics-aware spatio-temporal decomposer and a selective reprogrammed language model.  Experiment results confirm that the proposed framework unlocks the capabilities of PLMs to capture fine-grained spatio-temporal dynamics and achieves better performance than existing methods.

**Strengths:**

1. The paper is well-written and easy to follow.
2. The paper introduces a unique approach to enable PLMs to handle spatio-temporal data by using a physics-aware decomposer to disentangle complex spatio-temporal dynamics into components with rich physical semantics.
3. Extensive experiments are conducted to validate the effectiveness of the proposal.

**Weaknesses:**

1. The proposed decomposer is not clearly described. Since the decompostion is a common tool in time series analysis, there lacks a discussion of why the decomposed components are physics-aware.
2. The usage of the reconstruction matrix is ambiguous. Further ablation studies are needed.
3. Most Transformer-based baselines use 96 or longer input lengths. The paper only provides experimental results with an input length of 48, and the experiment under increasing the input length is missing.

**Questions:**

Please refer to the weaknesses.

---

> ### Author Response · Authors · 2024-11-20
>
> We sincerely thank the reviewer for the valuable feedback. We group related points in Weaknesses (W) and Questions (Q) as follows.
> ### W1: The proposed decomposer is not clearly described. Since the decomposition is a common tool in time series analysis, there lacks a discussion of why the decomposed components are physics aware.
>
>   **[Response]** Thank you for your valuable feedback. Our decomposer fundamentally differs from traditional decomposition methods. Below we provide a more detailed explanation of our decomposer and its physics-aware characteristics.
>
>   Our evolutionary decomposition method builds upon the principles of **Dynamic Mode Decomposition** and **Koopman operator theory**, both of which are rooted in capturing the intrinsic dynamics of a system through a linear approximation of its non-linear behavior. By analyzing the dominant modes, our evolutionary decomposition can uncover patterns that are both interpretable and consistent with the underlying physical mechanisms, providing valuable insights for PLM that can improve predictive accuracy.
>
>   As many studies have demonstrated, our DMD-based evolutionary decomposition extracts both spatial coherence and temporal dynamics, which represent key phenomena such as propagating waves, oscillatory behaviors, and decay patterns that are closely linked to dominant physical processes [1, 2]. The spatial modes obtained through decomposition reveal coherent structures inherent to the physical system. Importantly, the spatial modes retain the relationships dictated by the governing physical laws, ensuring the decomposition aligns with the true behavior of the system. For instance, it has been shown that DMD can uncover vortex shedding and periodic oscillations in fluid dynamics [2]. These patterns align closely with the Navier-Stokes equations, capturing the fundamental physical dynamics governing fluid motion [3]. Furthermore, each mode identified by evolutionary decomposition is associated with a temporal eigenvalue, which encodes the growth, decay, or oscillatory behavior of that mode. These eigenvalues reflect critical physical phenomena such as natural frequencies, damping rates, and stability margins.
>
>   Unlike PCA or Fourier decomposition, which primarily extract statistical patterns, our approach ensures the temporal evolution and spatial structure are inherently connected, providing a physically meaningful framework for analyzing and predicting system behavior. Our method ensures that each mode is tied to the system’s dynamics, making it particularly well-suited for spatio-temporal datasets governed by physical laws.
>
>    We will add these discussions in the revised manuscript. Thank you again for your thoughtful feedback!
>
> [1] Rowley C W, Mezić I, Bagheri S, et al. Spectral analysis of nonlinear flows[J]. Journal of fluid mechanics, 2009, 641: 115-127.
>
> [2] Tu J H. Dynamic mode decomposition: Theory and applications[D]. Princeton University, 2013.
>
> [3] Kutz J N, Brunton S L, Brunton B W, et al. Dynamic mode decomposition: data-driven modeling of complex systems[M]. Society for Industrial and Applied Mathematics, 2016.
>
> ### W2: The usage of the reconstruction matrix is ambiguous. Further ablation studies are needed.
>
>   **[Response]** Thank you for your valuable feedback, and we are sorry for any lack of clarity regarding the usage of the reconstruction matrix.
>
> As described in lines [198–208], reconstruction plays a pivotal role in DMD-related techniques [1,2], ensuring that the extracted modes faithfully capture the system's underlying dynamics while eliminating redundant noise. In our approach, we enhance this process by prioritizing the most dominant modes, enabling a more precise representation of key spatio-temporal patterns.
>
>   We agree that additional ablation studies are crucial to further clarify the impact of the reconstruction matrix on model performance. To address this, we conducted experiments to systematically evaluate its role and assess its influence on the overall results, following the setting of the ablation study in the paper.
>
>   w/o reconstruction: RePST with only dynamic components (X_dyn) as X_dec
>
>   w/o dynamics: RePST with only reconstruction data (X_rec) as X_dec
>
>
>
> ||||||||
> |-|-|-|-|-|-|-|
> ||Air Quality||Solar Energy||Beijing Taxi||
> |FULL SHOT|MAE|RMSE|MAE|RMSE|MAE|RMSE|
> |w/o reconstruction|30.32|45.34|3.76|5.98|19.08|35.43|
> |w/o dynamics|27.98|42.33|3.60|5.53|16.44|29.87|
> |RePST|26.20|39.37|3.27|5.12|15.13|25.44|
>
>
>   As can be seen in the table above, the model performance declines obviously when we remove either dynamic components (X_dyn) or reconstruction data(X_rec). The ablation studies confirm that the reconstruction matrix is essential for maintaining high performance, alongside the dynamic’s component. The results also demonstrate the necessity of integrating both components to fully realize the advantages of the RePST framework.
>
>   Thank you once again for bringing this to our attention!

---

> > ### Author Response · Authors · 2024-11-20
> >
> > ### W3: Most Transformer-based baselines use 96 or longer input lengths. The paper only provides experimental results with an input length of 48, and the experiment under increasing the input length is missing.
> >
> >   **[Response]** Thanks for your suggestion.
> >
> >   In our work, we initially focused on a relatively short input length, following common evaluation protocols in existing spatio-temporal forecasting research [4,5,6,7,8].
> >
> >   To address your concern, we conducted additional experiments using an input and prediction lengths of 96 across four datasets. As can be seen in the following table, our model, namely RePST, consistently outperformed strong baseline models, showcasing the robustness of our approach across varying input lengths. These results highlight RePST’s ability to effectively handle longer input sequences while maintaining superior performance.
> >
> >   Thank you again for your insightful suggestion, which has helped us strengthen our study.
> >
> >
> >
> > |96-96|Air Quality||Solar Energy||NYC Bike (inflow)||NYC Bike (outflow)||
> > |-|-|-|-|-|-|-|-|-|
> > ||MAE|RMSE|MAE|RMSE|MAE|RMSE|MAE|RMSE|
> > |iTransformer|50.92|72.95|4.58|9.91|4.45|10.34|4.45|10.33|
> > |PatchTST|44.22|66.67|4.24|8.56|4.47|10.88|4.48|10.88|
> > |STID|38.92|58.91|4.61|8.93|4.46|10.38|4.43|10.32|
> > |GWNet|40.47|60.44|OOM|OOM|7.59|16.43|7.53|16.28|
> > |RePST|34.34|54.38|4.02|8.05|4.4|9.95|4.39|9.87|
> >
> >
> >
> >
> >   [4] Yuan Y, Shao C, Ding J, et al. Spatio-temporal few-shot learning via diffusive neural network generation[C]//The Twelfth International Conference on Learning Representations. 2024.
> >
> >   [5] Cini A, Marisca I, Zambon D, et al. Taming local effects in graph-based spatiotemporal forecasting[J]. Advances in Neural Information Processing Systems, 2024, 36.
> >
> >   [6] Xia Y, Liang Y, Wen H, et al. Deciphering spatio-temporal graph forecasting: A causal lens and treatment[J]. Advances in Neural Information Processing Systems, 2024, 36.
> >
> >   [7] Li Z, Xia L, Xu Y, et al. GPT-ST: generative pre-training of spatio-temporal graph neural networks[J]. Advances in Neural Information Processing Systems, 2024, 36.
> >
> >   [8] Li Y, Yu R, Shahabi C, et al. Diffusion Convolutional Recurrent Neural Network: Data-Driven Traffic Forecasting[C]//International Conference on Learning Representations. 2018.
> >
> >
> >
> >
> >
> >   Finally, we sincerely thank you for your thoughtful feedback and valuable comments on our work. If possible, we kindly ask you to reconsider the ratings. Please do not hesitate to reach out with any remaining questions or concerns, as we are fully committed to clarifying any aspects of the paper that may require further elaboration.

---

> ### Author Response · Authors · 2024-11-25
> **Kind Reminder and Request for Reviewers' Feedback**
>
> Dear Reviewer EE3c,
>
> We sincerely appreciate your time and effort in reviewing our manuscript and offering valuable suggestions. As the author-reviewer discussion phase is drawing to a close, we would like to confirm whether our responses have effectively addressed your concerns. We provided detailed responses to your concerns a few days ago, and we hope they have adequately addressed your issues. If you require further clarification or have any additional concerns, please do not hesitate to contact us. We are more than willing to continue our communication with you.
>
> Best,
>
> Authors

---

> > ### Comment · Reviewer_EE3c · 2024-11-26
> >
> > Thanks for the detailed response, which basically addresses my concerns, and I would like to improve my score.

---

> > > ### Author Response · Authors · 2024-12-03
> > > **Sincere Gratitude from Authors**
> > >
> > > We are happy that our responses have effectively addressed your concerns. We would like to express our sincerest gratitude once again for taking the time to review our paper and provide us with such detailed and invaluable suggestions!

---

### Official Review · Reviewer_Xa2y · 2024-10-28

**Soundness:** 3
**Presentation:** 3
**Contribution:** 2
**Rating:** 5
**Confidence:** 4

**Summary:**

The paper proposes REPST, a novel framework for spatio-temporal forecasting that leverages Pre-trained Language Models (PLMs), traditionally used for text, by adapting them for numerical time series analysis. Recognizing the limitations of PLMs in modeling complex spatio-temporal correlations, REPST introduces two key components: a physics-aware decomposer that breaks down spatially correlated time series into interpretable sub-components, enhancing PLM understanding through a divide-and-conquer strategy; and a selective discrete reprogramming scheme that expands the spatio-temporal vocabulary, minimizing information loss and enriching PLM representations. Experiments on real-world datasets show that REPST outperforms twelve existing methods, demonstrating strong performance, especially in data-scarce settings, and unlocking the potential of PLMs for spatio-temporal tasks.

**Strengths:**

* I think the authors’ approach to leveraging PLMs for spatio-temporal forecasting is innovative, especially considering the usual challenges these models face with numerical time series. By adapting PLMs for spatio-temporal data, they explore an intriguing direction that could have broader implications for forecasting tasks in various fields.

* The proposed REPST framework’s integration of the physics-aware decomposer and selective discrete reprogramming is a creative attempt to enrich PLM comprehension of complex spatio-temporal patterns. This combination appears to facilitate a more structured understanding of the input data, which is promising for zero-shot and few-shot learning.

* The forecasting results presented in Table 1 are interesting, showing that REPST outperforms state-of-the-art baselines, especially in data-scarce scenarios. This suggests that the framework could be beneficial in practical applications where limited data is a significant constraint.

**Weaknesses:**

* I’m not entirely convinced by the claim of “physics-aware” decomposition. The use of the Dynamic Mode Decomposition (DMD) model seems more like pattern extraction from the input data X rather than incorporating actual physical laws. DMD is primarily designed for linear systems and is mainly focused on capturing patterns. If the authors are labeling it as “physics-aware,” I wonder why they didn’t opt for eigenvectors or simpler methods like PCA, which could also provide interpretable components. This makes me question whether the use of DMD here truly justifies the physics-informed label. Please clarify your reasoning behind using DMD over other methods like PCA or eigenvectors. Additionally, please provide more evidence or examples of how your decomposition method incorporates physical principles beyond pattern extraction.

* The terminology of “reprogramming” feels overstated to me. Typically, reprogramming would imply substantial changes to the PLM’s architecture or layers. Based on Figure 2, however, it seems that the base architecture of ChatGPT2 is not significantly modified, apart from the input transformation to align with spatio-temporal dynamics. I would appreciate a clearer explanation of what changes were made to the PLM, and whether these modifications involved retraining or fine-tuning beyond input alignment. Please provide a more detailed explanation of the changes made to the PLM architecture, if any, and clarify whether any retraining or fine-tuning was involved beyond input alignment.

* The results in Table 1 are intriguing, but I wonder if there is an inconsistency in how the baseline models were evaluated. It appears that REPST benefits from augmented data generated through the DMD process, while the baselines might have used only the original data. I believe a fair comparison would require running the baselines on the same augmented data to better understand the performance gap. Please clarify whether the baseline models were evaluated using the same augmented data as REPST, and if not, provide results of baselines run on the augmented data for a fair comparison.

* I find the claims about reasoning and generalizability improvements through PLM usage to be a bit unclear. The authors emphasize generalization, demonstrated by zero-shot performance improvements, but I don’t see a convincing demonstration of enhanced reasoning capabilities, particularly in spatial dimensions. If the improved reasoning is attributed to the DMD-based decomposition, it seems weak, as DMD primarily enhances temporal embedding rather than spatial reasoning. Please provide specific examples or analyses that demonstrate enhanced reasoning capabilities, particularly in spatial dimensions. Also, please clarify how DMD contributes to spatial reasoning, if at all.

**Questions:**

* Is the physics-aware component primarily derived from DMD modes? If so, how does this differ from other decomposition methods like PCA or eigenvectors, which can also capture patterns in data?

* How does DMD, which primarily captures temporal embeddings, contribute to enhancing spatial information? I’m still unclear about how DMD facilitates better spatial representation in the context of spatio-temporal dynamics.

* Could autoencoders, which also offer non-linear embeddings, serve as an alternative to the DMD for capturing dynamic information? Would such embeddings also be considered physics-aware in this context as you also have some augmentation from the data?

* How exactly does the authors’ approach to “reprogramming” the PLM differ from simply changing input structures? Were there any modifications to the PLM architecture or retraining steps involved?

* Have the baseline models been tested with the same augmented data as REPST? If not, how would the results compare under such conditions?

* How do the authors substantiate their claim of improved reasoning capabilities in the model, especially in terms of spatial reasoning? Is there specific evidence beyond improved zero-shot performance?

---

> ### Author Response · Authors · 2024-11-20
>
> We sincerely thank the reviewer for the valuable feedback. We group related points in Weaknesses (W) and Questions (Q) as follows.
>
> ### W1 & Q1: I wonder why they didn’t opt for eigenvectors or simpler methods like PCA, which could also provide interpretable components. This makes me question whether the use of DMD here truly justifies the physics-informed label. Please clarify your reasoning behind using DMD over other methods like PCA or eigenvectors. Additionally, please provide more evidence or examples of how your decomposition method incorporates physical principles beyond pattern extraction. Is the physics-aware component primarily derived from DMD modes? If so, how does this differ from other decomposition methods like PCA or eigenvectors, which can also capture patterns in data?
>
>   **[Response]** Thank you for your detailed feedback and raising these important questions! Below we provide a point-to-point response for your comments.
>
>   **Physics-Aware Justification**
>
>   Although DMD is designed to extract dynamic modes directly from observational data, numerous studies have demonstrated its capability to identify propagating waves, oscillatory behaviors, and decay patterns, which are strongly associated with dominant physical phenomena [1,2,3]. For example, research has shown that DMD can reveal vortex shedding patterns and periodic oscillations in fluid dynamics [1]. These patterns are closely aligned with the Navier-Stokes equations, reflecting the intrinsic physical dynamics of fluid motion [2,3]. For this reason, DMD is particularly suitable for capturing physics-aware patterns from spatio-temporal data, such as traffic flow and air pollution, which exhibit similar regularities with those governing fluid dynamics. For instance, STDEN [4] and AirPhyNet [5] model the dynamics of traffic flow and air pollution as a continuous diffusion process via differential equations inspired by fluid dynamics modeling.
>
>   **Why DMD Over PCA or Eigenvectors?**
>
>   Unlike DMD, PCA or eigenvector-based methods are purely statistical tools that do not inherently account for the underlying physical behavior of a system. By analyzing the dominant modes, DMD can uncover patterns that are both interpretable and consistent with the underlying physical mechanisms, providing valuable insights for PLM that can improve predictive accuracy.
>
> To validate our choice, we first conducted experiments by replacing the physical decomposer with PCA. This results in a noticeable decline in model performance, suggesting that PCA’s inability to benefit PLM in our context. Furthermore, when we randomly initialized the pretrained weights of the PLM, the impact on performance was minor, further indicating that PCA could not effectively leverage the pretrained knowledge. These findings underscore the superiority of DMD in generating physically consistent interpretations and unlocking pretrained model knowledge.
>
>   PCA_RePST: RePST with PCA as decomposer
>
>   PCA_random_RePST: RePST with PCA as decomposer and randomly initialize the weights of the PLM
>
> ||||||||||||
> |-|-|-|-|-|-|-|-|-|-|-|
> ||Air Quality ||Solar Energy  ||NYC Bike (inflow)  ||NYC Bike (outflow)  ||Beijing Taxi  ||
> |Few-Shot|MAE|RMSE|MAE|RMSE|MAE|RMSE|MAE|RMSE|MAE|RMSE|
> |PCA_RePST|37.49|51.66|4.61|9.74|7.51|15.33|7.81|15.16|29.68|49.57|
> |PCA_random_RePST|38.04|54.32|4.67|10.84|7.55|15.78|7.82|15.23|30.04|50.79|
> |RePST|33.57|47.3|3.65|6.74|5.29|12.11|5.66|12.85|26.85|45.88|

---

> > ### Author Response · Authors · 2024-11-20
> >
> > Furthermore, to verify the effectiveness of the knowledge extracted by the physical decomposer, we conducted experiments that apply the decomposed data to two state-of-the-art spatio-temporal forecasting models, i.e., STID and GWNet. The results showed only marginal improvement, indicating that even advanced spatio-temporal forecasting models struggle to effectively utilize the physical knowledge. This demonstrates that the knowledge extracted by the physical decomposer is not simply a set of fine-grained features but represents a unique form of information that cannot be easily leveraged by all models.
> >
> >   GWNet_decomposer: GWNet using decomposed features
> >
> >   STID_decomposer: STID with decomposed features
> >
> >   RePST_random: randomly initialize the weights of the PLM in our model
> >
> > ||||||||||
> > |-|-|-|-|-|-|-|-|-|
> > ||Air Quality||Solar Energy||NYC Bike (inflow)||NYC Bike (outflow)||
> > |Few-Shot|MAE|RMSE|MAE|RMSE|MAE|RMSE|MAE|RMSE|
> > |GWNet_decomposer|35.81|51.94|8.94|11.02|11.84|20.67|10.96|19.48|
> > |GWNet|36.26|54.88|9.10|11.87|12.55|21.97|12.68|22.27|
> > |STID_decomposer|42.44|59.68|4.34|8.75|8.07|16.04|8.04|16.55|
> > |STID|43.21|61.07|4.89|9.41|8.94|16.34|8.88|15.77|
> > |RePST_random|40.12|56.27|5.21|9.32|7.83|16.41|6.81|15.82|
> > |RePST|33.57|47.3|3.65|6.74|5.29|12.11|5.66|12.85|
> >
> >   In summary, DMD demonstrates a clear advantage over simpler methods such as PCA or eigenvector decomposition by integrating spatial and temporal dynamics through interpretable modes aligned with physical phenomena. Our additional experimental results further emphasize its ability to enhance PLM's performance by capturing meaningful, physics-aware patterns rather than merely extracting fine-grained statistical features.
> >
> > Thank you again for your thoughtful comments, which has helped us strengthen our paper. We will include these discussions into the next version of our manuscript.
> >
> >   [1] Rowley C W, Mezić I, Bagheri S, et al. Spectral analysis of nonlinear flows[J]. Journal of fluid mechanics, 2009, 641: 115-127.
> >
> >   [2] Tu J H. Dynamic mode decomposition: Theory and applications[D]. Princeton University, 2013.
> >
> >   [3] Yu Y, Liu D, Wang B, et al. Analysis of Global and Key PM2. 5 Dynamic Mode Decomposition Based on the Koopman Method[J]. Atmosphere, 2024, 15(9): 1091.
> >
> >   [4] Hettige, Kethmi Hirushini, Jiahao Ji, Shili Xiang, Cheng Long, Gao Cong, and Jingyuan Wang. "AirPhyNet: Harnessing Physics-Guided Neural Networks for Air Quality Prediction." In The Twelfth International Conference on Learning Representations.
> >
> >   [5] Ji, Jiahao, Jingyuan Wang, Zhe Jiang, Jiawei Jiang, and Hu Zhang. "STDEN: Towards physics-guided neural networks for traffic flow prediction." In Proceedings of the AAAI Conference on Artificial Intelligence, vol. 36, no. 4, pp. 4048-4056. 2022.

---

> > ### Comment · Reviewer_Xa2y · 2024-11-25
> >
> > Thank you for replying to my reviews and including these experiments. My questions about the physical-aware models are still unresolved. However, I think you are mixing the concepts of physical laws and spatial-temporal dependencies. In another word, you are mixing the concepts of physics-informed machine learning and representation learning. Those fluid motions and examples you mentioned, do have underlying physical equations to explain the system dynamics. Those physics-aware or physics-informed methods generally use data-driven methods to approach some parameters of the physic formula (usually some hard-to-solve PDE equations). In your case, those decomposers are just some representation learning to tell the spatial and temporal information, which is the abuse of terminology physics-aware or physics-informed field.

---

> ### Author Response · Authors · 2024-11-20
>
> ### W2 &Q4: The terminology of “reprogramming” feels overstated to me. How exactly does the authors’ approach to “reprogramming” the PLM differ from simply changing input structures? Were there any modifications to the PLM architecture or retraining steps involved?
>
>   **[Response]** Sorry for the confusion. The term “reprogramming,” as employed here and in prior studies [6] [7], does not refer to architectural changes or substantial retraining of the PLM. Instead, it describes the process of **adapting or reformatting input data** to align with the model’s existing architecture and capabilities. Rather than modifying the model’s internal structure or layers, reprogramming capitalizes on the inherent representational power of pretrained models by transforming input data to fit their original design.
>   We hope this clarification addresses any misunderstanding regarding the use of the term “reprogramming.” Our approach aims to maximize the utility of pretrained models by adapting inputs and outputs, rather than altering the underlying architecture, making it an efficient and powerful method for cross-modal adaptation.
>   Thank you again for your valuable feedback, which has helped us enhance the clarity and depth of our manuscript.
>
>   [6] Jin M, Wang S, Ma L, et al. Time-LLM: Time Series Forecasting by Reprogramming Large Language Models[C]//The Twelfth International Conference on Learning Representations.
>
>   [7] Yang C H H, Tsai Y Y, Chen P Y. Voice2series: Reprogramming acoustic models for time series classification[C]//International conference on machine learning. PMLR, 2021: 11808-11819.
>
> ### W3 & Q5: I wonder if there is an inconsistency in how the baseline models were evaluated. Please clarify whether the baseline models were evaluated using the same augmented data as REPST, and if not, provide results of baselines run on the augmented data for a fair comparison. Have the baseline models been tested with the same augmented data as REPST? If not, how would the results compare under such conditions?
>   **[Response]** Thanks for your constructive suggestion!
>   We acknowledge the importance of evaluating baseline models under consistent conditions. The time series forecasting models (e.g., PatchTST, iTransformer) were originally designed to process multivariate time series data structured as (B, L, C), where B is the batch size, L is the sequence length, and C is the number of variables. Adapting these models to spatio-temporal forecasting tasks—where the data is structured as (B, L, N, C'), with N denotes the number of spatial nodes and C' is the decomposed feature dimension—requires treating N as variable dimension (i.e., the dimension C in time series data (B, L, C)). This adaptation overlooks decomposed features (C'), preventing these models from fully leveraging the augmented data.
>   To evaluate the impact of augmented data on these time series forecasting models, we explored an alternative adaptation by reshaping the spatio-temporal data into (B*N, L, C'), allowing the models to access the augmented features. The results showed that due to the lack of explicit spatial modeling capabilities, these models performed even poorly when provided with augmented data.
>
> ||||||||||
> |-|-|-|-|-|-|-|-|-|
> ||Air Quality||Solar Energy||NYC Bike (inflow)||NYC Bike (outflow)||
> |Few-Shot|MAE|RMSE|MAE|RMSE|MAE|RMSE|MAE|RMSE|
> |PatchTST|35.76 |53.80|4.65|7.82|7.03 |15.32|6.88 |14.84|
> |PatchTST_augmented|44.44|65.58|13.29|37.68|20.08|79.01|19.85|77.34|
> |iTransformer|35.59 |52.95|4.74| 8.27|8.23 |16.21|7.46 |15.68|
> |iTransformer_augmented|46.83|70.45|12.83|36.51|21.18|82.45|21.82|82.41|
>
>
>   For state-of-the-art spatio-temporal forecasting models, we conducted additional experiments using the same augmented data. While some improvements were observed, the gains were limited. These models struggled to effectively utilize the augmented data, likely because they fail to interpret and harness the physics-aware knowledge encapsulated in the augmented features.
>
> ||||||||||
> |-|-|-|-|-|-|-|-|-|
> ||Air Quality||Solar Energy||NYC Bike (inflow)||NYC Bike (outflow)||
> |Few-Shot|MAE|RMSE|MAE|RMSE|MAE|RMSE|MAE|RMSE|
> |GWNet|36.26|54.88|9.10|11.87|12.55|21.97|12.68|22.27|
> |GWNet_augmented|35.81|51.94|8.94|11.02|11.84|20.67|10.96|19.48|
> |STID|43.21|61.07|4.89|9.41|8.94|16.34|8.88|15.77|
> |STID_augmented|42.44|59.68|4.34|8.75|8.07|16.04|8.04|16.55|
>
>
>   The above results confirm that the physics-aware augmented data provides unique benefits when integrated with PLMs in our RePST framework. In contrast, other models that solely rely on fine-grained features fail to effectively utilize this information. We hope this explanation resolves your concerns and underscores the strengths of our method.
>
> Thank you once again for your valuable feedback, which is instrumental in enhancing our analysis!

---

> > ### Author Response · Authors · 2024-11-20
> >
> > ### W4 & Q2 & Q6: I find the claims about reasoning and generalizability improvements through PLM usage to be a bit unclear. How does DMD, which primarily captures temporal embeddings, contribute to enhancing spatial information? I’m still unclear about how DMD facilitates better spatial representation in the context of spatio-temporal dynamics. How do the authors substantiate their claim of improved reasoning capabilities in the model, especially in terms of spatial reasoning? Is there specific evidence beyond improved zero-shot performance?
> >
> >   **[Response]** We appreciate your insightful comments, and sorry for the confusion.
> >
> >   First, in our work, reasoning refers to the ability to make predictions by comprehending both spatial and temporal contexts. While zero-shot performance improvements demonstrate generalization, we argue that they also reflect enhanced reasoning capabilities. In zero-shot setting, the model encounters spatial regions or domains it has not seen during training. The model's predictive performance solely relies on its inherent reasoning capabilities and its capacity to infer patterns from prior spatio-temporal knowledge.
> >
> >   Second, our DMD-based decomposer inherently captures spatial correlations within spatio-temporal data by leveraging the co-evolving nature of spatial nodes. Specifically, the input matrix X, where rows represent spatial nodes and columns represent time steps, implicitly embeds spatial relationships through the correlated dynamics of the nodes.  During decomposition, spatial nodes exhibiting similar temporal behaviors (e.g., traffic flow on adjacent roads) are naturally grouped into the same dynamic mode, i.e., dominant global patterns that summarize the behavior of correlated spatial nodes. These patterns reflect how the spatio-temporal system evolves over time. Furthermore, DMD does not require prior spatial knowledge (e.g., spatial adjacency matrices), making it a flexible approach for datasets with implicit spatial relationships.
> >
> >   Third, we further conducted empirical studies to validate the contribution of DMD to spatial reasoning. In specific, we decomposed each node in the dataset independently and then concatenated them together, rather than decomposing the entire spatio-temporal system. By doing so, we can eliminate the impact of DMD on spatial dimension. We observe an obvious decrease in the model performance, indicating that the decomposer extract spatio-temporal knowledge from the system rather than simple temporal embeddings.
> >
> >   separate_RePST: RePST with decomposer for each node independently
> >
> > ||||||||||||
> > |-|-|-|-|-|-|-|-|-|-|-|
> > ||Air Quality||Solar Energy||NYC Bike (inflow)||NYC Bike (outflow)||Beijing Taxi||
> > |FEW SHOT|MAE|RMSE|MAE|RMSE|MAE|RMSE|MAE|RMSE|MAE|RMSE|
> > |separate_RePST|35.81|53.66|10.66|18.03|8.44|16.32|8.67|17.03|30.67|56.43|
> > |RePST|33.57|47.3|3.65|6.74|5.29|12.11|5.66|12.85|26.85|45.88|
> >
> >   We hope these additions clarify the model’s spatial reasoning capabilities and the specific role of DMD.
> >
> >   ### Q3: Could autoencoders, which also offer non-linear embeddings, serve as an alternative to the DMD for capturing dynamic information? Would such embeddings also be considered physics-aware in this context as you also have some augmentation from the data?
> >
> >   **[Response]** Thank you for your valuable feedback. We apologize for any confusion caused.
> >
> >   Autoencoders are primarily designed for data compression. The embeddings they generate lack explicit ties to physical laws or system dynamics. The interpretability of such embeddings remains limited compared to DMD-based decomposition, which provides a direct mapping to the underlying dynamics of the system.
> >
> >   Therefore, while autoencoders can capture nonlinearities, their embeddings do not inherently possess the same physical significance as DMD modes and eigenvalues, making them less suitable for physics-driven analysis.
> >   We further conducted experiments by replacing our decomposer by autoencoder (RePST_autoencoder). As can be seen in the table below, the model performance decreases obviously, indicating that such autoencoder methods cannot provide embeddings with physical interpretability, which can be comprehended by PLMs.
> >
> > ||||||||||
> > |-|-|-|-|-|-|-|-|-|
> > ||Air Quality||Solar Energy||NYC Bike (inflow)||NYC Bike (outflow)||
> > |Few-Shot|MAE|RMSE|MAE|RMSE|MAE|RMSE|MAE|RMSE|
> > |RePST_autoencoder|38.17|52.27|5.04|8.82|6.43 |15.05|6.21|14.65|
> > |RePST|33.57|47.3|3.65|6.74|5.29|12.11|5.66|12.85|
> >
> >
> >
> >
> > Thank you so much for your insightful feedback and helpful suggestions regarding our work. If it’s possible, we kindly request you to reconsider the ratings. Please feel free to reach out if you have more questions, as we are eager to provide any additional explanations needed.

---

> > ### Comment · Reviewer_Xa2y · 2024-11-25
> >
> > Thank you for relying on my W3 & Q5 reviews and experiments. How about the performance for those time series data if they have C and C' features, or actually RePST just uses C' features?

---

> ### Author Response · Authors · 2024-11-25
> **Kind Reminder and Request for Reviewers' Feedback**
>
> Dear Reviewer Xa2y,
>
> We sincerely appreciate your time and effort in reviewing our manuscript and offering valuable suggestions. As the author-reviewer discussion phase is drawing to a close, we would like to confirm whether our responses have effectively addressed your concerns. We provided detailed responses to your concerns a few days ago, and we hope they have adequately addressed your issues. If you require further clarification or have any additional concerns, please do not hesitate to contact us. We are more than willing to continue our communication with you.
>
> Best,
>
> Authors

---

> ### Author Response · Authors · 2024-12-03
> **Response to Reviewer  Xa2y**
>
> Thanks for your prompt reply, which helps us further understand your concerns. We are pleased to hear that some of your questions have been addressed. Please find a point-to-point response to your new comments below.
>
> [Q1]
>
> > However, I think you are mixing the concepts of physical laws and spatial-temporal dependencies. In another word, you are mixing the concepts of physics-informed machine learning and representation learning. Those fluid motions and examples you mentioned, do have underlying physical equations to explain the system dynamics. Those physics-aware or physics-informed methods generally use data-driven methods to approach some parameters of the physic formula (usually some hard-to-solve PDE equations). In your case, those decomposers are just some representations learning to tell the spatial and temporal information, which is the abuse of terminology physics-aware or physics-informed field.
>
> **[Response]**
>
> We first highlight the distinction between **physics-aware** and **physics-informed** approaches [1,2,3].
>
> • **Physics-informed** methods explicitly integrate physical equations (e.g., PDEs) into the modeling process, often to learn specific parameters or solve otherwise intractable equations. Examples include Physics-Informed Neural Networks (PINNs) that embed governing equations directly into the loss function.
>
> • **Physics-aware** methods, on the other hand, do not explicitly incorporate physical equations but are designed to recognize, extract, or align with the underlying physical behaviors or dynamics from data. They use domain knowledge to ensure the results are physically meaningful without directly solving the equations. For example, PhAST [1] learns physics-aware embeddings without explicitly incorporating physical equations. But instead, it relies on **features that align closely with physical principles**. This approach is **similar to DMD**, which approximates the dynamics of complex systems in a data-driven manner **without explicitly solving physical equations**.
>
> So DMD is **not physics-informed** in the strict sense because it does not incorporate governing equations or constraints directly into its formulation. Instead, its physics-aware nature lies in its ability to **discover and align with physical patterns from data**, making it highly valuable in contexts where explicit equations are unknown or impractical to solve.
>
> Moreover, we respectfully invite the reviewers take a look at the Section 3 and 3.4 of a recent survey paper [4], where Koopman Theory methods (e.g., DMD) are also regarded as physical knowledge, although such methods do not explicitly incorporate physical equations. Referring to DMD as 'physics-aware' is therefore **reasonable** and **consistent** with the distinction between data-driven methods that identify physically consistent patterns and those that directly embed physical laws.
>
> We appreciate your thoughtful comments and agree that revising our terminology can help avoid misunderstandings. Based on your feedback, we propose adding more details in related work to further discuss the difference between our method and physics-informed learning. Thank you for helping us refine our explanation.
>
>
>
> [1] Duval A, Miret S, Bengio Y, et al. PhAST: Physics-Aware, Scalable, and Task-Specific GNNs for Accelerated Catalyst Design[J]. Journal of Machine Learning Research, 2024.
>
> [2] Gurbuz S Z. Physics-Aware Machine Learning for Dynamic, Data-Driven Radar Target Recognition[C]//International Conference on Dynamic Data Driven Applications Systems. Cham: Springer Nature Switzerland, 2022: 114-122.
>
> [3] Zheng Y, Zhan J, He S, et al. Curricular contrastive regularization for physics-aware single image dehazing[C]//Proceedings of the IEEE/CVF conference on computer vision and pattern recognition. 2023: 5785-5794.
>
> [4] Meng C, Seo S, Cao D, et al. When physics meets machine learning: A survey of physics-informed machine learning[J]. arXiv preprint arXiv:2203.16797, 2022.
>
> [Q2]
>
> > How about the performance for those time series data if they have C and C' features, or actually RePST just uses C' features?
>
> **[Response]** Thanks for your insightful feedback! We are willing to conduct experiments on RePST when just uses C'. Similarly, we reshape the input data as (B*N, L, C',1) like other time series models. As can be seen in the table below, the model performance decreases. It demonstrated that only leveraging C' is not effective for RePST. However, compared to other time series models like PatchTST and iTransformer, RePST still achieves better performance even when spatial information is not modeled (using only C' ).
>
> ||||||||||
> |-|-|-|-|-|-|-|-|-|
> ||Air Quality||Solar Energy||NYC Bike (inflow)||NYC Bike(outflow)||
> |FEW SHOT|MAE|RMSE|MAE|RMSE|MAE|RMSE|MAE|RMSE|
> |C'_RePST|34.73|52.66|8.32|15.05|7.82|16.03|7.67|16.03|
> |RePST|33.57|47.3|3.65|6.74|5.29|12.11|5.66|12.85|
>
>
> Thanks again so much for your insightful feedback and helpful suggestions regarding our work!

---

### Official Review · Reviewer_i6Kn · 2024-11-02

**Soundness:** 3
**Presentation:** 3
**Contribution:** 2
**Rating:** 3
**Confidence:** 4

**Summary:**

The paper proposes REPST, a spatiotemporal prediction framework that enables PLM (Pre-trained Language Models) to understand complex spatiotemporal patterns through a reprogramming strategy based on physics-aware decomposition. This framework employs a physics-aware decomposer to decompose spatially correlated time series, enhancing the model's ability to comprehend the patterns. Additionally, the paper introduces a selective discrete reprogramming scheme, which projects spatiotemporal series into discrete representations.

**Strengths:**

1. This paper is the first to propose a physics-aware spatio-temporal decomposer.
2. The experimental datasets span the fields of traffic, solar energy, and air quality, offering good diversity.
3. REPST demonstrates strong performance under the parameters and datasets specified in the paper.

**Weaknesses:**

1. The statement that "the rich physical semantic information can boost the pretrained physical knowledge of PLMs" (Line 288-231) lacks experimental or theoretical evidence demonstrating that it was specifically the physical knowledge of PLMs that contributed to the results. It seems more likely that the physical methods applied simply extracted features that were more reasonable and of finer granularity.
2. The paper, i.e., abstract part, mentions that the physics-aware decomposer allows PLM to understand complex spatiotemporal dynamics using a divide-and-conquer strategy. How exactly does this relate to the divide-and-conquer approach?
3. The experimental setup is limited to a short prediction length, with no results for other prediction lengths.
4. The experimental results do not include standard deviations and there is no mention of random seeds used, making it difficult to assess the statistical significance and reproducibility of the results.
5. The complete numerical results for Figure 3 are not provided in a tabular format, which would have been helpful for detailed comparison.
6. The paper could benefit from more visualizations to better illustrate how the physics-aware decomposer enhances interpretability.
7. There are relatively few models compared when evaluating Zero-Shot performance.
8. There is a typo in the middle of Section A.5, specifically in the last line of page 17 (line 916).

**Questions:**

1. Does REPST perform well only for short prediction lengths?
2. Line 212: If the concatenation of two parts in $X_{dec}$ is necessary, it would be necessary to include this component in the ablation study to assess its impact. Additionally, sensitivity analysis should be conducted to evaluate the effect of the hyperparameter $\alpha$.
3. It is recommended to provide the detailed settings of the ablation experiments in the appendix, as the current text only gives a very brief overview of the approach.
4. In the ablation study, why does the performance of the Solar Energy dataset show relatively little impact when the pretrained model is not used, compared to the other two datasets?
5. Why is the dimension of E' in Figure 2 inconsistent with its description in the text?
6. Line 285: What model does "HI" represent in line 285?
7. Since the Koopman theory-based evolutionary matrix can describe the evolution of the system over time $t$, is it possible to use matrix $\mathcal{A}$ directly for prediction? If so, it is recommended to include it in the comparative experiments.

---

> ### Author Response · Authors · 2024-11-20
>
> We sincerely thank the reviewer for the valuable feedback. We group related points in Weaknesses (W) and Questions (Q) as follows.
>
> ### W1: The statement that "the rich physical semantic information can boost the pretrained physical knowledge of PLMs" (Line 288-231) lacks experimental or theoretical evidence demonstrating that it was specifically the physical knowledge of PLMs that contributed to the results. It seems more likely that the physical methods applied simply extracted features that were more reasonable and of finer granularity.
>
>   **[Response]** Thanks for your insightful comments.
>   We conducted experiments in our paper (Figure 5) to test the effectiveness of PLMs by replacing PLMs with trainable attention layers. To further verify the effectiveness of pre-trained knowledge in PLMs, we conducted additional experiments as follows:
>   First, we examined whether the decomposed features could benefit baseline models. Specifically, we fed the decomposed feature into two state-of-the-art spatio-temporal forecasting models, i.e., GWNet and STID. The results showed only limited performance improvements, suggesting that even advanced spatio-temporal forecasting models cannot fully exploit these decomposed physical features. This finding indicates that the observed performance boost in our model cannot be solely attributed to fine-grained feature engineering.
>   Second, we removed the pre-trained knowledge in the PLM by randomly initializing its weights. As can be seen in the following table, this led to a significant performance decline, highlighting the importance of pretrained knowledge in comprehending and utilizing the decomposed physical semantics.
>
>   GWNet_decomposer: GWNet using decomposed features
>
>   STID_decomposer: STID with decomposed features
>
>   RePST_random: randomly initialize the weights of the PLM in our model
>
> ||||||||||
> |-|-|-|-|-|-|-|-|-|
> ||Air Quality ||Solar Energy ||NYC Bike (inflow) ||NYC Bike (outflow) ||
> |Few-Shot|MAE|RMSE|MAE|RMSE|MAE|RMSE|MAE|RMSE|
> |GWNet_decomposer|35.81|51.94|8.94|11.02|11.84|20.67|10.96|19.48|
> |GWNet|36.26|54.88|9.10|11.87|12.55|21.97|12.68|22.27|
> |STID_decomposer|42.44|59.68|4.34|8.75|8.07|16.04|8.04|16.55|
> |STID|43.21|61.07|4.89|9.41|8.94|16.34|8.88|15.77|
> |RePST_random|40.12|56.27|5.21|9.32|7.83|16.41|6.81|15.82|
> |RePST|33.57|47.3|3.65|6.74|5.29|12.11|5.66|12.85|
>
>   In conclusion, these experiments collectively demonstrate that the pretrained knowledge embedded in PLMs is a critical factor driving the performance gains, rather than solely the extraction of finer-grained features through physical decomposition. We hope this clarifies our approach and addresses your concerns.
>
> ### W2: The paper, i.e., abstract part, mentions that the physics-aware decomposer allows PLM to understand complex spatiotemporal dynamics using a divide-and-conquer strategy. How exactly does this relate to the divide-and-conquer approach?
>
>   **[Response]** Sorry for the confusion. The “divide-and-conquer” approach mentioned in the abstract specifically refers to the role of the physics-aware decomposer in simplifying the modeling of complex spatio-temporal dynamics.
>   As explained in the paper (lines 65–66), spatio-temporal dynamical systems are significantly more complex and densely coupled than standard time series data, making it challenging to model them directly. To address this, the decomposer divides input signal into multiple sub-components, each characterized by distinct and interpretable spatio-temporal semantics. By isolating these components, the PLM can focus on comprehending and modeling specific patterns more effectively, rather than contending with the overwhelming complexity of the system.
>   This process embodies the essence of divide-and-conquer strategy, where a complex problem is broken down into manageable sub-problems, each of which can be addressed independently. In this way, the decomposer enables the PLM to concentrate on distinct aspects of the spatio-temporal dynamics, ultimately enhancing overall comprehension and performance.

---

> ### Author Response · Authors · 2024-11-20
>
> ### W3&Q1: The experimental setup is limited to a short prediction length, with no results for other prediction lengths. Does REPST perform well only for short prediction lengths?
>
>   **[Response]** Thank you for your valuable and constructive feedback. As is common in spatio-temporal forecasting research (e.g.,[1], [2], [3], [4], [5].), our experiments focused on relatively short prediction lengths to align with standard evaluation protocols in the field. However, we acknowledge the importance of exploring longer prediction horizons to provide a more comprehensive assessment of model performance. In response, we conducted additional experiments with a prediction length of 96 across four datasets (OOM: Out-Of-Memory).
>
> ||||||||||
> |-|-|-|-|-|-|-|-|-|
> ||Air Quality ||Solar Energy ||NYC Bike (inflow) ||NYC Bike (outflow) ||
> |Few-Shot|MAE|RMSE|MAE|RMSE|MAE|RMSE|MAE|RMSE|
> |iTransformer|50.92|72.95|4.58|9.91|4.45|10.34|4.45|10.33|
> |PatchTST|44.22|66.67|4.24|8.56|4.47|10.88|4.48|10.88|
> |STID|38.92|58.91|4.61|8.93|4.46|10.38|4.43|10.32|
> |GWNet|40.47|60.44|OOM|OOM|7.59|16.43|7.53|16.28|
> |RePST|34.34|54.38|4.02|8.05|4.4|9.95|4.39|9.87|
>
>   As can be seen, RePST consistently outperformed powerful baseline models even in long-term predictions, underscoring the robustness of our approach across varying prediction lengths.
>   We understand the need for broader experimental coverage and are committed to extending our evaluations to include all baselines and datasets for longer prediction lengths in the revised manuscript. We hope this addresses your concern, and we sincerely thank you again for your valuable suggestion.
>
>   [1] Yuan Y, Shao C, Ding J, et al. Spatio-temporal few-shot learning via diffusive neural network generation[C]//The Twelfth International Conference on Learning Representations. 2024.
>
> [2] Cini A, Marisca I, Zambon D, et al. Taming local effects in graph-based spatiotemporal forecasting[J]. Advances in Neural Information Processing Systems, 2024, 36.
>
>   [3] Xia Y, Liang Y, Wen H, et al. Deciphering spatio-temporal graph forecasting: A causal lens and treatment[J]. Advances in Neural Information Processing Systems, 2024, 36.
>
>   [4] Li Z, Xia L, Xu Y, et al. GPT-ST: generative pre-training of spatio-temporal graph neural networks[J]. Advances in Neural Information Processing Systems, 2024, 36.
>
>   [5] Li Y, Yu R, Shahabi C, et al. Diffusion Convolutional Recurrent Neural Network: Data-Driven Traffic Forecasting[C]//International Conference on Learning Representations. 2018.
>
>   [6] Wu Z, Pan S, Long G, et al. Graph wavenet for deep spatial-temporal graph modeling[C]//Proceedings of the 28th International Joint Conference on Artificial Intelligence. 2019: 1907-1913.
>
> ### W4: The experimental results do not include standard deviations and there is no mention of random seeds used, making it difficult to assess the statistical significance and reproducibility of the results.
>
>   **[Response]** Thank you for your thoughtful suggestion! We recognize the importance of statistical significance and reproducibility in evaluating experimental results.
>   To clarify, all reported performance metrics for RePST are averaged over three independent runs with different random seeds to ensure reliability in the results. While standard deviations are not currently included in the paper, we will incorporate these details in the future revision to enhance the statistical rigor of our work.
>   Furthermore, we have prepared the codebase for our experiments, to facilitate full reproducibility of our results. This code will be publicly released soon to ensure transparency and support further exploration by the community. Thank you again!

---

> > ### Author Response · Authors · 2024-11-20
> >
> > ### W5 & W7: The complete numerical results for Figure 3 are not provided in a tabular format, which would have been helpful for detailed comparison. There are relatively few models compared when evaluating Zero-Shot performance.
> >
> >   **[Response]** Thanks for your valuable feedback! We appreciate your suggestion and understand that presenting the numerical results for Figure 3 in a tabular format would facilitate a more detailed comparison.
> >   We provide the complete numerical results corresponding to Figure 3 below:
> >
> > |||||||||
> > |-|-|-|-|-|-|-|-|
> > |Category|RePST|TimesFM|OpenCity-plus|OpenCity Base|OpenCity Mini|FPT|Time-LLM|
> > |NYC -> CHI|2.03|9.07|6.39|3.61|6.38|12.56|10.32|
> > |CHI -> NYC|11.9|19.23|13.26|13.48|13.4|30.24|25.44|
> > |NYC -> Solar|5.53|9.81|10.37|10.3|10.38|22.36|18.04|
> > |Solar -> Air|31.86|38.62|37.34|45.44|48.71|68.44|OOT|
> > |CHI -> Solar|5.57|9.81|10.17|10.3|10.38|26.32|16.28|
> > |Solar -> CHI|3.96|9.07|6.39|3.61|6.38|15.44|OOT|
> >
> >   Regarding the limited number of models compared for zero-shot performance, this is primarily due to the current scarcity of open-source spatio-temporal forecasting models explicitly designed for zero-shot capabilities. To the best of our knowledge, the baselines we included represent the available state-of-the-art in this area. However, we acknowledge the importance of broader comparisons, so we conducted more zero-shot experiments on time series models (FPT, Time-LLM [7]). While these models may not be specifically designed for spatio-temporal zero-shot forecasting tasks, their inclusion will provide a more comprehensive context for evaluation.
> >   As can be seen in the table, both FPT and Time-LLM fall short in such spatio-temporal prediction tasks, primarily due to the lack of abilities for spatial modeling. Moreover, because of the huge computational cost of Time-LLM caused by large amount of spatial variables, we could only complete experiments on relatively small datasets (OOT for out of time). This further demonstrates the superiority of our model in handling spatio-temporal tasks compared to the time series models.
> >
> >   [7] Jin M, Wang S, Ma L, et al. Time-LLM: Time Series Forecasting by Reprogramming Large Language Models[C]//The Twelfth International Conference on Learning Representations.
> >
> > ### W6: The paper could benefit from more visualizations to better illustrate how the physics-aware decomposer enhances interpretability.
> >
> >   **[Response]** Thanks for your nice suggestion! We provide a case study for our decomposer and selected vocabulary for visualization.
> >   Noted that these selective word embeddings are encodings for word or word morphemes, which can be regarded as the smallest unit that makes up a word. To intuitive display, we artificially combine these small units into words. In our RePST, this part is completed by cross attention module, which can automatically match the spatio-temporal data with most relevant words. Specifically, to minimize the subjective impact, we decrease the number of our expanded spatio-temporal vocabulary as 100. Finally, we present a case study on real-world spatio-temporal datasets. We sample the lookback window of one single node in Air Quality and visualize the selected vocabulary learned by differentiable discrete reprogramming blocks. As shown in tables below, it can be clearly observed that words from the selected vocabulary can jointly describe the temporal trend along with the spatio-temporal pattern vividly, indicating the effectiveness of reprogramming spatio-temporal data into textual representations.
> >   Part of our selective vocabulary: { Annual, Begin, rival, Bloom, refreshed, Gained, Wide, Split, incre, imitation, extremely, rising, few, inserted, core, air, ight, slowdown, cause, dawn, soon, erie, rapid, recalls,  Develop, wer, St, Less, industry, move, icted, bad, regained, rain, atmosp, wind, unchanged, dusty,  itialized, intervals, Benefit, crease, spread, holding,  ...}
> >   We also observed the insightful interpretability of our framework. For temporal components, series trends are reflected with words like "increase". Words like "Annual", "rapid", and "unchanged" also show temporal patterns.
> > For spatial components, we get words for relative relations such as "move" and "spread". Furthermore, some words describe the pattern in certain scenarios vividly. "dusty", "rain" and "wind" represent a kind of phenomena which have strong relationship with air quality. The above results demonstrate that RePST effectively captures the characteristics of different scenarios and can be used for various downstream tasks.

---

> > > ### Author Response · Authors · 2024-11-20
> > >
> > > ### W8: Typos
> > >
> > >   **[Response]** Thanks for your kind feedback. We apologize for the typos, and we will correct them in the revised manuscript.
> > >
> > > ### Q2: Line 212: If the concatenation of two parts in X_dec is necessary, it would be necessary to include this component in the ablation study to assess its impact. Additionally, sensitivity analysis should be conducted to evaluate the effect of the hyperparameter α.
> > >
> > >   **[Response]** Thank you for your valuable feedback, and we apologize for any confusion caused.
> > >   We agree that it is important to assess the impact of the concatenation process in through an ablation study. To address your concern, we conducted additional experiments specifically examining the role of this reconstruction component, following the ablation study part in our paper. We conducted experiments, which only leverage dynamic components or reconstruction.
> > >
> > >   w/o reconstruction: RePST only uses reconstruction X_rec
> > >
> > >   w/o dynamics: RePST only uses dynamic components X_dyn
> > >
> > > ||||||||
> > > |-|-|-|-|-|-|-|
> > > ||Air Quality ||Solar Energy ||Beijing Taxi ||
> > > |Full-Shot|MAE|RMSE|MAE|RMSE|MAE|RMSE|
> > > |w/o reconstruction|30.32|45.34|3.76|5.98|19.08|35.43|
> > > |w/o dynamics|27.98|42.33|3.60|5.53|16.44|29.87|
> > > |RePST|26.20|39.37|3.27|5.12|15.13|25.44|
> > >
> > >   As can be seen in the table, the performance of all the datasets decreases obviously when we remove either the reconstruction or dynamic components, indicating that both make an effect to the whole framework.
> > >
> > >   Furthermore, we conducted a sensitivity analysis to evaluate the influence of the number of selected modes in α. We show the performance of Air Quality dataset with different number of selected modes. As can be seen in the table below, the performance is not good enough when there are few selected modes, which cannot describe the whole system well. When there are more than three modes, the increasement of the amount of α is not necessary. It may cause performance decline due to the introduction of noise.
> > >
> > > ||||
> > > |-|-|-|
> > > ||Air Quality||
> > > |#selected-modes|MAE|RMSE|
> > > |1|37.38|57.72|
> > > |2|35.24|53.26|
> > > |3|34.31|50.93|
> > > |4|33.57|47.30|
> > > |5|33.61|47.42|
> > > |6|33.63|47.48|
> > >
> > > ### Q3: It is recommended to provide the detailed settings of the ablation experiments in the appendix, as the current text only gives a very brief overview of the approach.
> > >   **[Response]** Thanks for your valuable suggestion! We provide the detailed settings of the ablation study here for reference and will include these details in the revised version of our paper.
> > >   - r/p PLM: We replace the PLM backbone with a single trainable ramdomly-initialized multi-head attention layer, to test the effect of PLMs in RePST, following the setting of [8]. [8] conducted a series of ablation study to test the effect of PLMs in time series tasks.
> > >   - r/p Decomposition: To test the effect our physics-aware spatio-temporal decomposer, we replace it with a transformer encoder. Rather than extracting spatio-temporal knowledge with physical interpretability, we apply a one-layer transformer encoder to model spatio-temporal correlations.
> > >   - w/o expanded vocabulary: We remove our selective spatio-temporal vocabulary to test its contribution to the whole framework. We apply a linear layer to map the whole vocabulary to a relatively small amount, following the setting of Time-LLM [6].
> > >
> > >   [8] Tan M, Merrill M A, Gupta V, et al. Are Language Models Actually Useful for Time Series Forecasting? [C]//The Thirty-eighth Annual Conference on Neural Information Processing Systems.
> > >
> > > ### Q4: In the ablation study, why does the performance of the Solar Energy dataset show relatively little impact when the pretrained model is not used, compared to the other two datasets?
> > >
> > >   **[Response]** Thank you for your thoughtful feedback and for pointing out this observation. We apologize for any confusion caused.
> > >   The relatively smaller impact of not using the pretrained model on the Solar Energy dataset can be attributed to the simpler temporal patterns inherent in this dataset. Specifically, the Solar Energy dataset exhibits a straightforward and predictable pattern: solar energy generation drops to zero during nighttime due to the absence of sunlight. This periodic and highly regular behavior makes it significantly easier for models to learn and predict, even without leveraging the advanced capabilities of PLMs. In contrast, the other datasets contain more complex and less predictable patterns, which depend more heavily on the pretrained knowledge of PLMs to capture and generalize effectively.
> > >   We hope this explanation clarifies the observed differences, and we sincerely appreciate your valuable feedback.

---

> ### Author Response · Authors · 2024-11-20
>
> ### Q5: The inconsistent of E’ in Figure 2 and text
>
>   **[Response]** Thanks for your valuable feedback and we apologize for the confusion. We will fix the inconsistent expression in the revised manuscript.
>
> ### Q6: What model does "HI" represent in line 285?
>
>   **[Response]** Thank you for your comments, and we apologize for the confusion caused. "HI" refers to "Historical Inertia" (HI) [9], a baseline model designed to leverage the natural continuity of historical data without relying on trainable parameters. HI directly uses the historical data point closest to the prediction target within the input time series as the forecasted value.
>   HI capitalizes on the inherent persistence of historical patterns, making it a simple yet effective benchmark for time series forecasting tasks. While often overlooked, this method serves as a valuable reference point for assessing the performance of more complex models.
>
>   [9] Cui, Yue, Jiandong Xie, and Kai Zheng. "Historical inertia: A neglected but powerful baseline for long sequence time-series forecasting." *Proceedings of the 30th ACM international conference on information & knowledge management*. 2021.
>
> ### Q7: Since the Koopman theory-based evolutionary matrix can describe the evolution of the system over time t, is it possible to use matrix A directly for prediction? If so, it is recommended to include it in the comparative experiments.
>
>   **[Response]** Thanks for your useful suggestion!
>   Koopman theory-based evolution matrix is not commonly used for prediction in existing research. Instead, it is primarily employed as a tool for dynamic analysis rather than as a primary means of forecasting. Nonetheless, we are willing to conduct experiments that directly use the matrix A for prediction. As shown, such simple prediction methods fall significantly short of the performance achieved by our model RePST.
>
> ||||||||||||||||||
> |-|-|-|-|-|-|-|-|-|-|-|-|-|-|-|-|-|
> ||METR-LA||PEMS-BAY||Air Quality||Solar Energy||Beijing Taxi (inflow)||Beijing Taxi (outflow)||NYC Bike (inflow)||NYC Bike (outflow)||
> |Few-Shot|MAE|RMSE|MAE|RMSE|MAE|RMSE|MAE|RMSE|MAE|RMSE|MAE|RMSE|MAE|RMSE|MAE|RMSE|
> |prediction_from_A|9.11|15.64|20.97|29.31|81.38|105.19|6.93|12.24|145.85|239.27|144.18|237.11|18.05|33.53|16.91|30.94|
> |RePST|5.63|9.67|3.61|7.15|33.57|47.30|3.65|6.74|26.85|45.88|26.3|43.76|5.29|12.11|5.66|12.85|
>
> We sincerely thank you for your valuable feedback and constructive suggestions on our work. If possible, we kindly ask you to reconsider the ratings. Should there be any further questions or concerns, we would be more than happy to address them in detail.

---

> ### Author Response · Authors · 2024-11-25
> **Kind Reminder and Request for Reviewers' Feedback**
>
> Dear Reviewer i6Kn,
>
> We sincerely appreciate your time and effort in reviewing our manuscript and offering valuable suggestions. As the author-reviewer discussion phase is drawing to a close, we would like to confirm whether our responses have effectively addressed your concerns. We provided detailed responses to your concerns a few days ago, and we hope they have adequately addressed your issues. If you require further clarification or have any additional concerns, please do not hesitate to contact us. We are more than willing to continue our communication with you.
>
> Best,
>
> Authors

---

> > ### Comment · Reviewer_i6Kn · 2024-11-25
> > **Thank you for your response.**
> >
> > I have carefully reviewed the author’s response. While some of my questions have been addressed, several key concerns remain:
> >
> > 1. Unpromising experiments:
> > The experimental results are not impressive, especially considering the diverse prediction lengths involved in the forecasting tasks. Although results for long-term forecasting baselines have been provided, only a subset has been reported.
> > Additionally, I am puzzled by the performance discrepancy between iTransformer and PathTST. According to Table 10 in Liu et al. (2024), the performance rankings are reversed.
> >
> > 2. Rebuttal experimentation:
> > The inclusion of *numerous* additional experiments during the rebuttal stage suggests that the initial submission was neither comprehensive nor adequate in its experimental scope.
> >
> > 3. Over-claiming and concept clarity:
> > The paper tends to over-claim its contributions. Important concepts such as reprogramming and divide-and-conquer remain ambiguous despite being crucial to the paper’s framework. This lack of clarity contributes to the paper’s complexity and readability issues.
> > The notion of “interpretability” is frequently mentioned, yet it is not convincingly demonstrated how this is achieved within the context of the work.
> >
> > 4. Intuition of the physics-aware component:
> > As mentioned in my first weakness and also noted by Reviewer Xa2y, the intuition and motivation behind the incorporation of the physically aware component are not sufficiently explained. The author’s response to my question W1, which involved conducting experiments by ablating this component, did not convincingly demonstrate its significance or promise.
> >
> > Thus, I'll keep my rating score.

---

> > > ### Author Response · Authors · 2024-12-03
> > > **Response to Reviewer i6KN-Part 1**
> > >
> > > Thanks for your reply. Please find a point-to-point response to your new comments below.
> > >
> > > > [Q1-1] Unpromising experiments: The experimental results are not impressive, especially considering the diverse prediction lengths involved in the forecasting tasks. Although results for long-term forecasting baselines have been provided, only a subset has been reported.
> > >
> > > **[Response]** Thanks for your question.
> > >
> > > First, we would like to clarify that **existing studies on spatio-temporal forecasting (STF) [1, 2, 3, 4] focus on short-term scenarios**. For example, [1] and [2] pay attention to 24 prediction lengths, while [3], and [4] aim to predict future states within next 12 time steps. This is because most practical STF applications (e.g., traffic management) relies on short-term prediction results for enabling immediate decision-making.
> > >
> > > Second, **our work does not claim to address long-sequence forecasting as a primary objective**. Instead, our research centers on enhancing predictive performance in data-scarce scenarios. This is validated through extensive few-shot and zero-shot experiments in the manuscript, demonstrating the effectiveness of our model, a point that has been acknowledged positively by Reviewer **LuDs and** **Xa2y**.
> > >
> > > Third, in response to the reviewer's request, **we have conducted additional experiments on long-sequence forecasting, even though it is uncommon in spatio-temporal forecasting research.** The results show that our model achieves up to **11.7% performance improvement** over baselines on four datasets. Additionally, our model demonstrates a significant **performance enhancement of around 30% in average** in short-term **few-shot** and **zero-shot** forecasting, which is the main focus of this field. These improvements highlight the superiority of our approach, and we believe they substantiate the significance of our contributions.
> > >
> > >
> > > > [Q1-2] Additionally, I am puzzled by the performance discrepancy between iTransformer and PathTST. According to Table 10 in Liu et al. (2024), the performance rankings are reversed.
> > >
> > > **[Response]** We would like to clarify that our experiments were conducted under **the few-shot scenario**, which only involves limited data for model training. This setting significantly differs from **the full-data training setting** used to produce the results presented in Table 10 of Liu et al. (2024) (if Liu et al. (2024) refers to iTransformer). The difference in experimental setup naturally leads to variations in performance rankings between models. Moreover, we used a popular time series and spatio-temporal forecasting library, i.e., BasicTS [5], for baseline experiments to ensure the consistency and reproducibility of the results.
> > >
> > >
> > >
> > > [1] Cini A, Marisca I, Zambon D, et al. Taming local effects in graph-based spatiotemporal forecasting[J]. Advances in Neural Information Processing Systems, 2024, 36.
> > >
> > > [2] Xia Y, Liang Y, Wen H, et al. Deciphering spatio-temporal graph forecasting: A causal lens and treatment[J]. Advances in Neural Information Processing Systems, 2024, 36.
> > >
> > > [3] Li Z, Xia L, Xu Y, et al. GPT-ST: generative pre-training of spatio-temporal graph neural networks[J]. Advances in Neural Information Processing Systems, 2024, 36.
> > >
> > > [4] Li Y, Yu R, Shahabi C, et al. Diffusion Convolutional Recurrent Neural Network: Data-Driven Traffic Forecasting[C]//International Conference on Learning Representations. 2018.
> > >
> > > [5] Shao Z, Wang F, Xu Y, et al. Exploring progress in multivariate time series forecasting: Comprehensive benchmarking and heterogeneity analysis[J]. IEEE Transactions on Knowledge and Data Engineering, 2024.

---

> > > > ### Author Response · Authors · 2024-12-03
> > > > **Response to Reviewer i6KN-Part 2**
> > > >
> > > > > [Q2]: Rebuttal experimentation: The inclusion of *numerous* additional experiments during the rebuttal stage suggests that the initial submission was neither comprehensive nor adequate in its experimental scope.
> > > >
> > > > **[Response]** The additional experiments provided during the rebuttal stage, e.g., long term forecasting, more zeros-shot baselines, and direct prediction using evolution matrix, **are not necessary to the core contributions of our work**, as explained in previous response. These experiments are not intended to alter or redefine the original conclusions of the paper. Instead, they serve to further reinforce and supplement our findings, addressing specific reviewer's concerns. **We believe the initial submission already included a comprehensive and sufficient experimental scope to support the primary claims of the paper, i.e., few-shot and zero-shot forecasting.** The additional experiments were conducted to enhance clarity and provide further validation, not to address any gaps in the original submission.

---

> ### Author Response · Authors · 2024-12-03
> **Response to Reviewer i6KN-Part 3**
>
> > [Q3-1] Over-claiming and concept clarity: The paper tends to over-claim its contributions. Important concepts such as reprogramming and divide-and-conquer remain ambiguous despite being crucial to the paper’s framework. This lack of clarity contributes to the paper’s complexity and readability issues.
>
> **[Response]** The concepts of “reprogramming” [6,7] and “divide-and-conquer” [8] are **widely recognized** in the field. Our use of these terms aligns with their conventional meanings, and we do not claim to redefine or expand them beyond their established scope. Therefore, we respectfully **disagree** with the assertion that our work over-claims the contributions or makes any ambiguous statements on these two concepts. These concepts were build upon existing studies, and we ensured their usage was consistent with prior literature.
>
> [6] Jin M, Wang S, Ma L, et al. Time-LLM: Time Series Forecasting by Reprogramming Large Language Models[C]//The Twelfth International Conference on Learning Representations.
>
> [7] Yang C H H, Tsai Y Y, Chen P Y. Voice2series: Reprogramming acoustic models for time series classification[C]//International conference on machine learning. PMLR, 2021: 11808-11819.
>
> [8] Deng J, Chen X, Jiang R, et al. Disentangling Structured Components: Towards Adaptive, Interpretable and Scalable Time Series Forecasting[J]. IEEE Transactions on Knowledge and Data Engineering, 2024.
>
>
> > [Q3-2] The notion of “interpretability” is frequently mentioned, yet it is not convincingly demonstrated how this is achieved within the context of the work.
>
> > [Q4-1] Intuition of the physics-aware component: As mentioned in my first weakness and also noted by Reviewer Xa2y, the intuition and motivation behind the incorporation of the physically aware component are not sufficiently explained.
>
> **[Response]**
>
> **Physics-aware** methods do not explicitly incorporate physical equations but are designed to recognize, extract, or align with the underlying physical behaviors or dynamics from data [1,2,3]. They use domain knowledge to ensure the results are physically meaningful without directly solving the equations.  For example, PhAST [1] learns physics-aware embeddings without explicitly incorporating physical equations. But instead, it relies on **features that align closely with physical principles**. This approach is similar to DMD, which approximates the dynamics of complex systems in a data-driven manner **without explicitly solving physical equations**.
>
> So DMD is **not physics-informed** in the strict sense because it does not incorporate governing equations or constraints directly into its formulation. Instead, its physics-aware nature lies in its ability to **discover and align with physical patterns from data**, making it highly valuable in contexts where explicit equations are unknown or impractical to solve.
>
> Moreover, methods like DMD are also regarded as physical knowledge in recent studies [4], although such methods do not explicitly incorporate physical equations. Calling DMD **physics-aware** is therefore **reasonable** and aligns with the distinction between data-driven methods that respect physical principles and those that directly embed physical laws.
>
> [1] Duval A, Miret S, Bengio Y, et al. PhAST: Physics-Aware, Scalable, and Task-Specific GNNs for Accelerated Catalyst Design[J]. Journal of Machine Learning Research, 2024.
>
> [2] Gurbuz S Z. Physics-Aware Machine Learning for Dynamic, Data-Driven Radar Target Recognition[C]//International Conference on Dynamic Data Driven Applications Systems. Cham: Springer Nature Switzerland, 2022: 114-122.
>
> [3] Zheng Y, Zhan J, He S, et al. Curricular contrastive regularization for physics-aware single image dehazing[C]//Proceedings of the IEEE/CVF conference on computer vision and pattern recognition. 2023: 5785-5794.
>
> [4] Meng C, Seo S, Cao D, et al. When physics meets machine learning: A survey of physics-informed machine learning[J]. arXiv preprint arXiv:2203.16797, 2022.
>
> > [Q4-1] The author’s response to my question W1, which involved conducting experiments by ablating this component, did not convincingly demonstrate its significance or promise.
>
> **[Response]** We conducted detailed experiments by removing the pre-trained knowledge in the PLM through random weight initialization, resulting in a performance degradation of up to **29.49%**. Additionally, as reported in the ablation study section of our manuscript, replacing our decomposer with a transformer encoder led to a performance drop of up to **24.87%**. These experiments underscore the critical role and significance of our decomposer.

---

### Official Review · Reviewer_LuDs · 2024-11-04

**Soundness:** 3
**Presentation:** 3
**Contribution:** 4
**Rating:** 8
**Confidence:** 2

**Summary:**

This paper provides a novel method of using pre-trained large language models (PLMs) for spatiotemporal forecasting. The method, REPST, is a 3-step process:
1. Using the Spatio-Temporal Decomposer to extract "evolutionary signals" that summarize the system dynamics and encode these signals into patches
2. Using the Reprogrammed Language Model to transform the signal patch embeddings into the semantic space of the PLM such that the PLM can output the final latent representations
3. Using the Projection Layer to obtain the predictions
The paper evaluates REPST on 6 real-world datasets and demonstrates SOTA performance even with few-shot, zero-shot learning.

The overall contribution of the paper is providing a framework with dynamic mode decomposition that allows the usage of PLMs for accurate spatiotemporal modelling.

**Strengths:**

The paper is well-written in preparing the reader with sufficient background knowledge to fully understand REPST. Specifically, the appendix includes a section on Koopman Theory to justify the assumption of treating spatiotemporal data as a linear dynamic system. Section A.5. also describes the evolutionary decomposition step in detail, clearing questions readers might have about the method. At the same time, the paper provides very recent literature such as Koopa by Liu et al., 2024.

The overall framework is also creative. By decomposing the input into its most important signals and transforming them into the semantic latent space, the power of PLMs can be harnessed for prediction.

The ablation study also helps readers understand the effectiveness of the separate components in REPST.

The question being asked in this paper is very important since LLMs are rapidly developing and REPST bridges the progress from LLMs to spatiotemporal forecasting.

**Weaknesses:**

* **Writing clarity**:
    * Some parts of the paper use vague descriptions. For instance, from Line 226-231, it could be better to be more specific in what the "reprogramming" does, and how "rich physical semantic information can boost the pretrained physical knowledge of PLMs".
    >To enrich spatio-temporal semantics and enable more comprehensive modeling of hidden spatio-temporal physical relationships, as well as unlocking the reasoning capabilities of PLMs, we further reprogram the components into the textual embedding place via an expanded spatio-temporal vocabulary. When handling textual-based components, the rich physical semantic information can boost the pretrained physical knowledge of PLMs, resulting in an adequate modeling of the hidden physical interactions between disentangled components.
    * At lines 93-94, 467-468, and 537, the paper mentions that the spatio-temporal vocabulary enables PLMs to focus on relationships among 3D geometric space, but it is not further justified or explained in the main text. It is unclear why.
    * In many spatiotemporal datasets, $\mathbf{X}$ is not two-dimensional but three-dimensional. For instance, in Line 830, the paper mentions that the Air Quality dataset has 6 indicators, so $\mathbf{X} \in \mathbb{R}^{35 \times 24 \times 6}$.
    * CHI Bike dataset not referenced and included in Table 3.

* **Typos**:
    * At Line 44-45, GPT-3 is not by Bai et al., 2018
    * At Line 40-41, it may be better to quote DCRNN and STGCN
    * At Line 50, the FPT acronym is not spelled out
    * At Line 156 SVD not spelled out
    * At Line 195-196 and line 204-205 A.5. >> (see section A.5.)

**Questions:**

* Since this Selective Reprogrammed Language Model of REPST produces the Top-K word embeddings from the pre-trained vocabulary of the PLM, what are the Top-K words that the signal patch embeddings are being mapped to? This semantic relationship would make the model more interpretable.
* In terms of the experimental setup for traffic prediction, why is the prediction horizon 24, and not {3, 6, 12} like the baseline methods? This would allow for a more rigorous comparison in prediction performance. Additionally, for Figure 4, multiple prediction horizons are tested for the other datasets, but no data is provided on METR-LA and PEMS-BAY. Similarly, for Figure 5, only 3 datasets are shown, and the additional results are not found in the Appendix.
* What is the Projection Layer? Is it a fully connected layer? How does it project $\mathbf{Z}\_{text} \rightarrow \hat{\mathbf{Y}}$
* How is REPST transferred to different datasets with different numbers of nodes? E.g. how are the learnable matrices transferrable from Solar with 137 nodes to Air Quality with 35 nodes? Why is a new dataset "CHI Bike" introduced for zero-shot?
* At Line 213 what is $L_P$?
* The authors should include anonymized code for reproducibility

---

> ### Author Response · Authors · 2024-11-20
>
> We sincerely thank the reviewer for the valuable feedback. We group related points in Weaknesses (W) and Questions (Q) as follows.
>
> ### W1: Some parts of the paper use vague descriptions. For instance, from Line 226-231, it could be better to be more specific in what the "reprogramming" does, and how "rich physical semantic information can boost the pretrained physical knowledge of PLMs".
>
>   **[Response]** Thanks for your time and valuable feedback! We apologize for the confusion.
>
> The term “reprogramming,” refers to the process of **adapting or reformatting input data** to align with the model’s existing architecture and capabilities [1,2].
>
> Our DMD-based decomposer inherently account for the underlying physical behavior of a system. For example, research has shown that DMD can reveal vortex shedding patterns and periodic oscillations in fluid dynamics [3]. These patterns are closely aligned with the Navier-Stokes equations, reflecting the intrinsic physical dynamics of fluid motion [4]. By analyzing the dominant modes, DMD can uncover patterns that are both interpretable and consistent with the underlying physical mechanisms, providing valuable insights for PLM that can improve predictive accuracy.
>
> Leveraging such decomposition, we can disentangle the complex spatio-temporal system into more interpretable components, which can be described in human-like manners. This clear semantic information is inherently easy for PLMs to comprehend, which can boost the related pretrained physical knowledge of PLMs. Additionally, as can be seen in Figure 5 of our paper (table below), our decomposer which extracts rich physical semantic information contributes a lot when compared to directly modeling the spatio-temporal correlations by transformer encoder.
>
> ||||||||
> |-|-|-|-|-|-|-|
> ||Air Quality||Solar Energy||Beijing Taxi||
> ||MAE|RMSE|MAE|RMSE|MAE|RMSE|
> |r/p Decomposition|31.94|46.68|4.04|6.25|20.14|37|
> |RePST|26.2|39.37|3.27|5.12|15.13|25.44|
>
>
> Thank you once again for your valuable feedback!
>
>   [1] Jin M, Wang S, Ma L, et al. Time-LLM: Time Series Forecasting by Reprogramming Large Language Models[C]//The Twelfth International Conference on Learning Representations.
>
>   [2] Yang C H H, Tsai Y Y, Chen P Y. Voice2series: Reprogramming acoustic models for time series classification[C]//International conference on machine learning. PMLR, 2021: 11808-11819.
>
>   [3] Rowley C W, Mezić I, Bagheri S, et al. Spectral analysis of nonlinear flows[J]. Journal of fluid mechanics, 2009, 641: 115-127.
>
>   [4] Yu Y, Liu D, Wang B, et al. Analysis of Global and Key PM2. 5 Dynamic Mode Decomposition Based on the Koopman Method[J]. Atmosphere, 2024, 15(9): 1091.
>
>
>
>   ### W2: At lines 93-94, 467-468, and 537, the paper mentions that the spatio-temporal vocabulary enables PLMs to focus on relationships among 3D geometric space, but it is not further justified or explained in the main text. It is unclear why.
>
>   **[Response]** Thanks for your time and valuable feedback! We apologize for the confusion.
>
>   As can be seen in the ablation study part (Figure 5), we conducted experiments to test the effectiveness of the spatio-temporal vocabulary by replacing our selective spatio-temporal vocabulary and utilizes the dense mapping function from [1]. In the table from Figure 5, represented below, the model performance decreases obviously, which demonstrates that this fused vocabulary failed to provide insights for PLMs to comprehend 3D relationships (2D space and 1D time axis). In contrast, our selective spatio-temporal vocabulary is equipped with spatial-related and temporal-related words, which can well describe the spatio-temporal correlations and make the spatio-temporal semantic information clearer for PLMs to comprehend, leading to more precise modeling and prediction.
>
> ||||||||
> |-|-|-|-|-|-|-|
> ||Air Quality||Solar Energy||Beijing Taxi||
> ||MAE|RMSE|MAE|RMSE|MAE|RMSE|
> |w/o Expanded Vocabulary|33.56|48.54|4.12|7.03|21.94|37.91|
> |RePST|26.2|39.37|3.27|5.12|15.13|25.44|
>
>
> Thank you again for your insightful suggestion!

---

> > ### Author Response · Authors · 2024-11-20
> >
> > ### W3 & W4: In many spatiotemporal datasets, X is not two-dimensional but three-dimensional. For instance, in Line 830, the paper mentions that the Air Quality dataset has 6 indicators, so X∈R35×24×6. CHI Bike dataset not referenced and included in Table 3.
> >
> >   **[Response]** Thanks for your time and valuable feedback! We apologize for the confusion.
> >
> >   For datasets with three dimensions, we separate the feature to conduct experiments. For example, we divided the NYC Bike dataset into inflow and outflow. For Air Quality dataset, we focus on its first indicator: PM2.5.
> >
> >   For CHI Bike dataset, we provide the reference [4] and details here. And we will add this information in the revised manuscript.
> >
> > ||||||
> > |-|-|-|-|-|
> > |CHI Bike |270|(6183,  883,1766)|30 min|Bike Service|
> >
> >
> >   [4] Jiang J, Han C, Jiang W, et al. LibCity: A Unified Library Towards Efficient and Comprehensive Urban Spatial-Temporal Prediction[J]. arXiv preprint arXiv:2304.14343, 2023.
> >
> >   Thanks again for your feedback. We will add more details for datasets in appendix.
> >
> >   ### Q1: Since this Selective Reprogrammed Language Model of REPST produces the Top-K word embeddings from the pre-trained vocabulary of the PLM, what are the Top-K words that the signal patch embeddings are being mapped to? This semantic relationship would make the model more interpretable.
> >
> >   **[Response]** Thanks for your nice suggestion! We provide a case study for our selected word embeddings for visualization.
> >
> >   Noted that these selective word embeddings are encodings for word or word morphemes, which can be regarded as the smallest unit that makes up a word. To intuitive display, we artificially combine these small units into words. In our RePST, this part is completed by cross attention module, which can automatically match the spatio-temporal data with most relevant words. Specifically, to minimize the subjective impact, we decrease the number of our expanded spatio-temporal vocabulary as 100. Finally, we present a case study on real-world spatio-temporal datasets. We sample the lookback window of one single node in Air Quality and visualize the selected vocabulary learned by differentiable discrete reprogramming blocks. As shown in tables below, it can be clearly observed that words from the selected vocabulary can jointly describe the temporal trend along with the spatio-temporal pattern vividly, indicating the effectiveness of reprogramming spatio-temporal data into textual representations.
> >
> >   Part of our selective vocabulary: {Annual, Begin, rival, Bloom, refreshed, Gained, Wide, Split, incre, imitation, extremely, rising, few, inserted, core, air, ight, slowdown, cause, dawn, soon, erie, rapid, recalls,  Develop, wer, St, Less, industry, move, icted, bad, regained, rain, atmosp, wind, unchanged, dusty, , itialized, intervals, Benefit, crease, spread, holding,  ...}
> >
> >   We also observed the insightful interpretability of our framework. For temporal components, series trends are reflected with words like "increase" Words like "Annual", "rapid", and "unchanged" also show temporal patterns.
> > For spatial components, we get words for relative relations such as "move" and "spread". Furthermore, some words describe the pattern in certain scenarios vividly. "dusty", "rain" and "wind" represent a kind of phenomena which have strong relationship with air quality. The above results demonstrate that RePST effectively captures the characteristics of different scenarios and can be used for various downstream tasks.

---

> > > ### Comment · Reviewer_LuDs · 2024-11-27
> > >
> > > * **W3 & W4**:  Thank you for the clarification. Does this mean REPST cannot be applied to multi-dimensional spatiotemporal data? If not, could you briefly describe how it can be extended?
> > >
> > > * **Q1**: This is an interesting result. Could you describe how you curated the spatiotemporal vocabulary?

---

> > ### Comment · Reviewer_LuDs · 2024-11-27
> >
> > Thank you for the detailed response.
> >
> > * **W1**: Thank you for the clarifications. I believe these changes should make the paper easier to understand. On the other hand, the paper claims that there is "pretrained physical knowledge" in PLMs, the authors should provide further evidence or references for this fact.
> > * **W2**: My confusion was in the paper mixing *2D space and 1D time* with *3D* geometric space. In the response, the authors discuss an experiment that is unclear to me:
> > >we conducted experiments to test the effectiveness of the spatio-temporal vocabulary by replacing our selective spatio-temporal vocabulary and utilizes the dense mapping function from [1].
> >
> > Are you replacing the spatiotemporal vocabulary + dense mapping from [1] with your own spatiotemporal vocabulary? The authors then state that
> > >this fused vocabulary failed to provide insights for PLMs to comprehend 3D relationships (2D space and 1D time axis)
> >
> > What is the fused vocabulary? Can you provide an example?
> >
> > Thanks again for the response

---

> ### Author Response · Authors · 2024-11-20
>
> ### Q2: In terms of the experimental setup for traffic prediction, why is the prediction horizon 24, and not {3, 6, 12} like the baseline methods? This would allow for a more rigorous comparison in prediction performance. Additionally, for Figure 4, multiple prediction horizons are tested for the other datasets, but no data is provided on METR-LA and PEMS-BAY. Similarly, for Figure 5, only 3 datasets are shown, and the additional results are not found in the Appendix.
>
>   **[Response]** Thank you for your valuable feedback and for highlighting the need for more clarity.
>
>   The prediction horizons {3, 6, 12} are only standard setting for METR-LA and PEMS-BAY. To show performance on a longer horizon, we set the prediction horizon as 24.  However, we are still willing to provide the **few-shot experiment results** with the prediction horizon 12 on METR-LA and PEMS-BAY, as follows:
>
> ||||||
> |-|-|-|-|-|
> ||METR-LA||PEMS-BAY||
> ||MAE|RMSE|MAE|RMSE|
> |GWNet|6.43|13.24|2.35|5.11|
> |STID|7.76|15.51|2.45|5.63|
> |iTransformer|5.21|12.66|2.34|5.65|
> |PatchTST|5.23|12.57|2.33|5.63|
> |RePST|4.78|9.67|2.26|4.86|
>
>
>   For Figure 4, we conducted experiments on Air Quality and NYC Bike to test multiple prediction horizons. We are willing to add experiments on METR-LA and PEMS-BAY under the same setting here.
>
> ||||||||||||
> |-|-|-|-|-|-|-|-|-|-|-|
> ||6||12||24||36||48||
> |METR-LA|MAE|RMSE|MAE|RMSE|MAE|RMSE|MAE|RMSE|MAE|RMSE|
> |STAEformer|3.59|9.04|3.75|9.84|4.02|10.26|5.12|10.83|4.55|11.25|
> |STID|3.55|8.92|3.56|9.81|4.78|11.63|4.95|10.99|4.47|11.59|
> |GWNet|3.84|8.69|4.14|9.72|4.19|10.57|4.07|11.08|4.71|11.76|
> |RePST|3.23|8.38|3.48|9.44|3.89|9.93|3.96|10.73|4.15|11.08|
>
>
> ||||||||||||
> |-|-|-|-|-|-|-|-|-|-|-|
> ||6||12||24||36||48||
> |PEMS-BAY|MAE|RMSE|MAE |RMSE|MAE|RMSE|MAE|RMSE|MAE|RMSE|
> |STAEformer|3.61|6.87|4.49|7.71|5.65|9.32|5.64|10.54|6.21|11.21|
> |STID|2.71|5.61|3.47|7.11|5.02|9.49|5.85|10.63|6.43|11.23|
> |GWNet|4.21|8.02|4.64|8.66|5.21|9.48|5.66|10.31|6.18|11.42|
> |RePST|2.68|5.44|3.33|6.84|4.79|9.06|5.35|9.94|5.99|10.85|
>
>
>   As can be seen in the table above, our RePST remains relatively stable and robust, demonstrating its efficacy in leveraging PLM to improve performance over longer-term predictions. For ablation studies in Figure 5, we are doing our best to conduct experiments on these two datasets. We will provide the results once we get them.
>
> We have added the additional results of **ablation studies on METR-LA and PEMS-BAY** here.
>
> ||||||
> |-|-|-|-|-|
> ||PEMS-BAY||METR-LA||
> ||MAE|RMSE|MAE|RMSE|
> |r/p PLM|3.82|7.25|6.44|11.64|
> |r/p Decomposition|2.54|5.25|4.11|7.92|
> |w/o Expanded Vocabulary|2.95|6.26|5.88|10.42|
> |RePST|1.92|4.33|3.63|7.34|
>
>
>   Thank you again for your insightful suggestion, which has helped us strengthen our study.
>
> ### Q3: What is the Projection Layer? Is it a fully connected layer? How does it project
>
>   **[Response]** Thank you for your valuable feedback!
>
>   The Projection Layer in RePST serves as a learnable mapping function that transforms the latent representations from the frozen PLM into the output space required for spatio-temporal forecasting. As provided in our code, the projection layer is implemented as a fully connected layer. It projects high-dimensional textual embeddings into numerical outputs that represent future spatio-temporal states. This layer bridges the gap between the PLM’s textual representations and the forecasting task, aligning the output dimensions and semantics with the requirements of spatio-temporal prediction.

---

> > ### Author Response · Authors · 2024-11-20
> >
> > ### Q4: How is REPST transferred to different datasets with different numbers of nodes? E.g. how are the learnable matrices transferrable from Solar with 137 nodes to Air Quality with 35 nodes? Why is a new dataset "CHI Bike" introduced for zero-shot?
> >
> >   **[Response]** Thank you for your valuable feedback and for highlighting the need for more clarity.
> >
> >   As you see in our code, RePST is inherently transferable to datasets with different numbers of nodes because its architecture and parameters are not directly tied to the number of nodes in the dataset. Unlike many graph-based or node-specific models, all learnable parameters in RePST’s layers are defined independently of the node structure.
> >
> >   To test the zero-shot performance in cross-region setting, we introduce CHI Bike dataset, which is from the same field as NYC Bike but different region.
> >
> >   Thanks again for your valuable feedback!
> >
> >
> > ### Q5: At Line 213 what is L_P?
> >
> >   **[Response]** Thanks for your valuable feedback, and we apologize for the inconsistent statements.
> >
> >   The L_p refers to patch length. We will fix it to T_p in our revised manuscript.
> >
> >
> > ### Q6: The authors should include anonymized code for reproducibility
> >
> >   **[Response]** Thanks you for your valuable feedback!
> >
> >   We have provided the codebase for our experiments, to facilitate full reproducibility of our results. This code will be publicly released soon to ensure transparency and support further exploration by the community.
> >
> > Overall, we are truly grateful for your constructive feedback and insightful comments on our work. If you think it is reasonable, we would appreciate it if you could consider revisiting the confidence. If there are still any open questions or uncertainties, we are more than willing to provide further explanations and address them thoroughly.

---

> ### Author Response · Authors · 2024-11-25
> **Kind Reminder and Request for Reviewers' Feedback**
>
> Dear Reviewer LuDs,
>
> We sincerely appreciate your time and effort in reviewing our manuscript and offering valuable suggestions. As the author-reviewer discussion phase is drawing to a close, we would like to confirm whether our responses have effectively addressed your concerns. We provided detailed responses to your concerns a few days ago, and we hope they have adequately addressed your issues. If you require further clarification or have any additional concerns, please do not hesitate to contact us. We are more than willing to continue our communication with you.
>
> Best,
>
> Authors

---

> ### Comment · Reviewer_LuDs · 2024-11-27
>
> * **Q2**: In the original baseline papers, the MAE and RMSE for prediction horizon 12 is much lower. For instance, after full-training, Graph WaveNet has MAE 3.53 on the METR-LA dataset, and MAE 1.95 on the PEMS-BAY dataset. This limits the generalizability of the proposed method.
>
> * **Q3**: Thank you for the response, I believe this should be added to the paper for better clarify.

---

> ### Author Response · Authors · 2024-12-03
> **Response to Reviewer LuDs**
>
> Thanks for your prompt reply, which helps us further understand your concerns. We are pleased to hear that some of your questions have been addressed. Please find a point-to-point response to your new comments below.
>
> [W1]
>
> **[Response]** Thanks for your insightful reply. We would like to add the following references for this statement.
>
> Li et al. [1] demonstrates that pretrained physical knowledge exists in PLMs (Pretrained Language Models) through a benchmark called VEC (Visual and Embodied Concepts), which tests the models’ understanding of visual and embodied physical concepts. The findings highlight that PLMs encode some level of physical knowledge.
>
> Peng et al. [2] designed tasks to evaluate whether PLMs organize entities by conceptual similarities and understand conceptual properties. Their research indicates that PLMs possess some level of conceptual knowledge, which is foundational to understanding physical properties and relationships.
>
> [1] Li L, Xu J, Dong Q, et al. Can Language Models Understand Physical Concepts?[C]//Proceedings of the 2023 Conference on Empirical Methods in Natural Language Processing. 2023: 11843-11861.
>
> [2] Peng H, Wang X, Hu S, et al. COPEN: Probing Conceptual Knowledge in Pre-trained Language Models[C]//Proceedings of the 2022 Conference on Empirical Methods in Natural Language Processing. 2022: 5015-5035.
>
>
> [W2]
>
> **[Response]** PLMs often face challenges related to the vast number of word embeddings in their vocabulary, leading to the risk of overwhelming and inefficient reprogramming. To address this issue, compressing the vocabulary is an effective solution. Previous studies have employed a fully connected layer to reduce the vocabulary size [3], which integrates the original semantic meanings of the embeddings during the compression process. To be specific, it utilizes a fully connected layer, whose input dimension is the number of pretrain word embeddings, and output dimension is targeting number of compressed vocabulary.
>
> [W3 & W4]
>
> **[Response]** Thanks for your valuable feedback. Given that the decomposed sub-components are treated as features, adding too many feature dimensions is computationally inefficient. Moreover, these features are highly correlated with the target variable. Therefore, we do not recommend applying RePST to multi-dimensional spatiotemporal data. If an extension is needed, the data can be split into single-dimensional components, as demonstrated in the paper.
>
> [Q1]
>
> **[Response]** Thanks for your valuable feedback! To convert GPT-2’s word embedding into a token, we first compute the similarity (e.g., cosine similarity) between the embedding vector from our vocabulary and each embedding in the vocabulary. Identify the index of the closest match. Then, use **GPT-2’s tokenizer** to decode this index back into its corresponding token.
>
> [Q2]
>
> **[Response]** Thanks for your insightful feedback! We would like to clarify that our experiments were conducted in a **few-shot scenario**, utilizing only a limited amount of data for model training. The primary contribution of our work lies in demonstrating the effectiveness of RePST in **data-scarce scenarios**.
>
> To address your concern, we have provided **full training** experiment results with an input and prediction length of 12 on the METR-LA and PEMS-BAY datasets. As shown in the table below, our model, RePST, performs **comparably to state-of-the-art baseline models**, highlighting its robustness across varying input lengths. These results underscore RePST’s capability to effectively handle shorter input sequences while maintaining competitive performance.
>
> ||||||
> |-|-|-|-|-|
> ||PEMS-BAY||METR-LA||
> |Full Training|MAE|RMSE|MAE|RMSE|
> |STAEformer|1.88|4.34|3.34|7.02|
> |GWNet|1.95|4.52|3.53|7.37|
> |STID|1.89|4.40|3.55|7.55|
> |RePST|1.88|4.33|3.34|7.12|
>
>
> Additionally, our model demonstrates a significant **performance enhancement of around 30% in average** in short-term **few-shot** and **zero-shot** forecasting, which is the **main focus** of this field. This improvement highlights the **superior generalization capabilities** of our approach, and we believe they substantiate the significance of our contributions.
>
> Thank you again for your insightful suggestion, which has helped us strengthen our study.

---

### Author Response · Authors · 2024-11-22
**General response to all reviewers**

We sincerely thank all the reviewers for their valuable time and detailed feedback, and we appreciate that almost all reviewers recognized the novelty and significance of our work. We have carefully revised the paper according to the comments, and the revised contents have been highlighted in BLUE. We also provide a detailed response to each comment below. Here we highlight our major revisions, and respond to each reviewer below. We hope our responses can properly address the reviewers' concerns.

- In response to the feedback from **Reviewer** **LuDs**, **i6KN**, **X2ay** and **EE3c**, we have conducted additional experiments (amounting to over **300** new results in total) and revised the paper accordingly, along with the relevant discussions.
- In response to feedback from **i6KN** and **EE3c**, we have expanded our experiments to include longer prediction horizon (96) on 4 datasets: Air Quality, Solar Energy, NYC Bike (inflow), NYC Bike (outflow). For comprehensive results, please refer to Appendices.
- In response to feedback from **Reviewer** **i6KN** and **X2ay**, we added in-depth explanations on why our DMD-based decomposer is physics-aware and suitable for spatio-temporal data compared to methods like PCA or autoencoders.
- Addressing the feedback from **i6KN, X2ay **and** EE3c**, we provided more ablation studies with finer granularity to clarify the contributions of the decomposer, reconstruction matrix, and selective vocabulary.

Additionally, our code has been made available to AC and the reviewers.

We hope these revisions and additional experiments address all concerns and further demonstrate the rigor and significance of our study. If there are any remaining questions, we would be happy to provide further clarifications. Thank you again for your valuable feedback, which has greatly helped us improve the quality of our work.

---

### Author Response · Authors · 2024-11-25
**Kind Reminder and Request for Reviewers' Feedback**

Dear Reviewers,

We sincerely appreciate your time and effort in reviewing our manuscript and offering valuable suggestions.
As the author-reviewer discussion phase is drawing to a close, we would like to confirm whether our responses have effectively addressed your concerns.
We provided detailed responses to your concerns a few days ago, and we hope they have adequately addressed your issues. If you require further clarification or have any additional concerns, please do not hesitate to contact us. We are more than willing to continue our communication with you.

Best,

Authors

---

### Meta-Review · Area_Chair_Q8TZ · 2024-12-16

**Metareview:**

The paper introduces REPST, a novel framework designed to adapt pre-trained language models (PLMs) for spatio-temporal forecasting tasks. REPST operates in three steps: (1) leveraging a physics-aware spatio-temporal decomposer (based on Koopman theory) to extract key system dynamics; (2) reprogramming signal patches into the PLM’s semantic space to unlock its reasoning capabilities; and (3) using a projection layer for final predictions. The framework is evaluated on six real-world datasets across diverse domains such as traffic, solar energy, and air quality, demonstrating state-of-the-art performance in zero-shot and few-shot settings.

Strengths:
+ Innovation: REPST’s reprogramming strategy creatively aligns spatio-temporal data with PLMs, exploring new ground in adapting language models for numerical forecasting.
+ Performance: The framework outperforms state-of-the-art methods on multiple datasets, excelling in low-data scenarios.
+ Theoretical Contributions: A detailed discussion of Koopman theory and its integration into the decomposition process is included, which helps ground the approach in established methods.
+ Ablation Studies: Effective ablation studies clarify the roles of different components in the REPST pipeline.
+ Broad Applicability: The method is tested across diverse domains (e.g., air quality, solar energy, and traffic datasets), demonstrating its versatility.

Weaknesses:
+ Clarity Issues: The “physics-aware” label for the decomposer is not well-justified, and the contribution of physical semantics is ambiguous. Descriptions of reprogramming and its contribution to reasoning capabilities lack specificity.
+ Experimental Gaps: Missing results for longer input sequences and varying prediction horizons. No standard deviations or random seeds are reported, raising reproducibility concerns. The baseline comparisons may be unfair if the baselines were not evaluated with augmented data, as REPST was.
+ Interpretability: Claims of improved reasoning (especially spatial reasoning) are not convincingly demonstrated with examples or visualizations. The role of the DMD-based decomposition in spatial representation is unclear.
+ Additional Concerns: Typos, inconsistent terminology, and unclear descriptions (e.g., “FPT,” “HI,” and figure dimensions) detract from the presentation. Details on the projection layer, reconstruction matrix, and transferability to datasets with varying node counts are insufficiently explained.

Some concerns have been addressed by the authors during the rebuttal period.

**Additional Comments On Reviewer Discussion:**

This is a borderline paper, with two positive and two negative ratings. Reviewer i6Kn pointed out several issues of the papers, including experimental setup, standard deviations, and few models compared when evaluating Zero-Shot performance. Authors tried to address these concerns, but failed to convince her/him. Some of the issues are also highlighted by Reviewer Xa2y. Ultimately, both reviewers keep their negative ratings unchanged.

---

### Decision · Program_Chairs · 2025-01-22

Reject